# Scaling Group Inference for Diverse and High-Resolution Generation

**Gaurav Parmar**[1]    **Or Patashnik**[2,3]    **Daniil Ostashev**[2]    **Kuan-Chieh Wang**[2]

**Kfir Aberman**[2]    **Srinivasa Narasimhan**[1]    **Jun-Yan Zhu**[1]

[1]Carnegie Mellon University    [2]Snap Research    [3]Tel Aviv University

## Abstract

Generative models typically sample outputs independently, and recent inference-time guidance and scaling algorithms focus on improving the quality of individual samples. However, in real-world applications, users are often presented with *a set* of multiple images (e.g., 4-8) for each prompt, where independent sampling tends to lead to redundant results, limiting user choices and hindering idea exploration. In this work, we introduce a scalable group inference method that improves both the diversity and quality of a group of samples. We formulate group inference as a quadratic integer assignment problem: candidate outputs are modeled as graph nodes, and a subset is selected to optimize sample quality (unary term) while maximizing group diversity (binary term). To substantially improve runtime efficiency, we progressively prune the candidate set using intermediate predictions, allowing our method to scale up to large candidate sets. Extensive experiments show that our method significantly improves group diversity and quality compared to independent sampling baselines and recent inference algorithms. Our framework generalizes across a wide range of tasks, including text-to-image, image-to-image, and image prompting, enabling generative models to treat multiple outputs as cohesive groups rather than independent samples.

## 1 Introduction

Recent advances in generative models, such as diffusion models, have driven significant efforts in inference-time guidance and scaling techniques (Ho & Salimans, 2021; Ma et al., 2025; Parmar et al., 2023). These methods effectively improve various aspects of output quality, such as alignment with text prompts or image aesthetics, and offer fine-grained controls over the output. However, much recent work focuses on enhancing the quality of *individual* samples generated in isolation.

Yet, in real-world applications, users are often shown a group of samples rather than just one. For example, many platforms (Midjourney, Inc., 2024; Adobe Inc., 2025) display a grid of four to eight images per prompt by default, a practice that offers users crucial benefits: more diverse choices regarding layout, lighting, and style, and new inspirations and ideas for prompt refinement and local edits. This creates a gap between current research, focused on independent samples, and the practical need for diverse, high-quality groups in content creation workflows. How can we close this gap? Beyond creative applications, grouped generation is also essential in numerous practical settings where humans, or downstream systems, require groups of outputs rather than single images. Examples include synthetic data generation, where multiple diverse outputs (including long-tail and rare cases) are necessary for robust training; design and engineering workflows that evaluate several plausible candidate solutions. In all of these scenarios, group quality and diversity need to be jointly optimized.

In this work, we propose a *scalable group inference* method to jointly improve the diversity and quality of a collection of generated samples. We formulate this task as a quadratic integer programming problem, representing output candidates as graph nodes. From a large set of $M$ candidates, we select a subset of size $K$ that maximizes a combination of individual sample quality, as a unary term, and group diversity as a binary term. However, a direct approach involves running the $T$-step

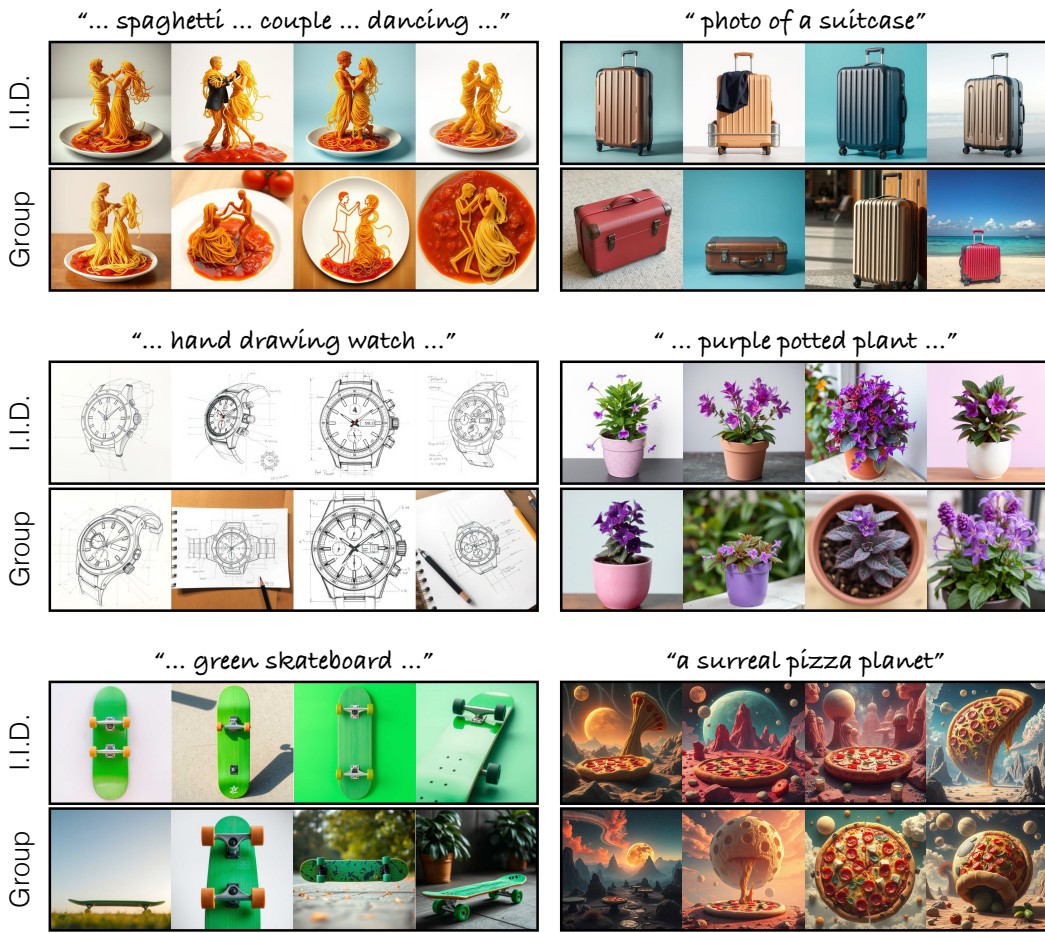

Figure 1: **Scalable Group Inference Results.** We show the advantage of our proposed group inference method over I.I.D. sampling. While I.I.D. sampling often yields repetitive results for the same prompt, our method generates a more diverse and high-quality collection of outputs.

denoising process for all $M$ candidates, resulting in an $\mathcal{O}(MT)$ complexity. This is computationally expensive for large $M$ and $T$ (e.g., $M = 128$, $T = 20$). To address this, we introduce an efficient progressive selection strategy that leverages intermediate predictions during denoising to iteratively prune the candidate set. This approach is grounded in the insight that these intermediate predictions of the final output, despite originating from a long denoising chain, serve as effective *previews* of the final image at each step (Figure 3). This approach reduces complexity to $\mathcal{O}(M + KT)$, where typically $K << M$, enabling us to scale up our group inference to handle large candidate sets.

Extensive experiments show that our group inference method significantly outperforms independent sampling baselines and recent algorithms across various generative tasks and modalities. Our method scales efficiently and produces more diverse and realistic outputs given the same compute budget. We further provide a comprehensive ablation study demonstrating the effectiveness of our design choices. In summary, our contributions are:

- We reformulate the generation of multiple outputs from a task of independent sampling to one of cohesive group inference. We are the first to formulate this as a Quadratic Integer Programming (QIP) problem that jointly optimizes for individual quality and group diversity.

- We introduce a progressive pruning algorithm that makes our QIP formulation practical and highly scalable. By leveraging intermediate $\mathbf{x}_0$ predictions as previews, this strategy reduces the computational complexity from $\mathcal{O}(MT)$ to $\mathcal{O}(M + KT)$.

- We demonstrate that our framework is well suited for inference-time scaling. For any given inference budget, progressive pruning allows for using a larger initial candidate set and obtains a higher quality and more diverse final output group.

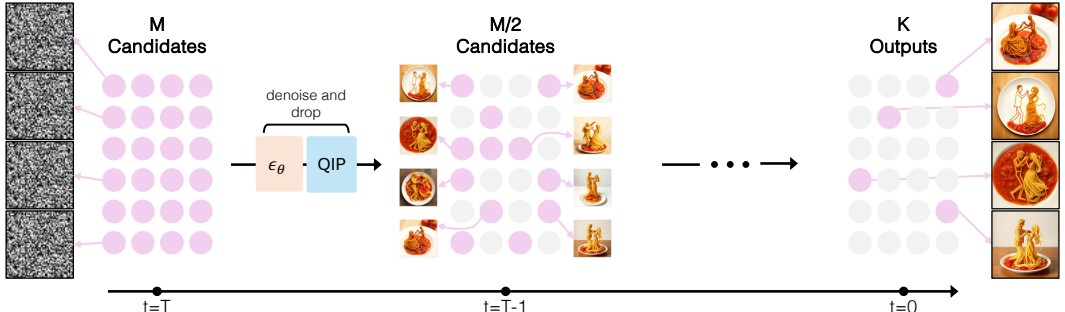

Figure 2: **Overview.** Given a large number of $M$ candidate noises, we gradually shrink the candidate set through iterative denoising and pruning steps. At each step, we first denoise the sample with the diffusion model $\epsilon_\theta$. We then compute the quality metric (unary term) and pairwise distances (binary term), and solve a quadratic integer programming (QIP) program to progressively prune the candidate set, yielding a final group of $K$ diverse and high-quality outputs.

- We provide extensive validation across multiple tasks, models, and metrics. Our method significantly outperforms baselines across both automated metrics and human preference studies.
- We show that our framework is flexible and capable of naturally incorporating different notions of diversity. This allows our scalable group inference process to be tailored to specific downstream applications or end-user preferences.

## 2 RELATED WORKS

**Diffusion models** synthesize high-quality samples through iterative denoising (Sohl-Dickstein et al., 2015; Ho et al., 2020; Song et al., 2020) and have shown success in text-to-image (Rombach et al., 2022; Podell et al., 2024; Saharia et al., 2022), video (Ho et al., 2022; Blattmann et al., 2023b;a) and 3D synthesis (Poole et al., 2022; Lin et al., 2023; Shi et al., 2023). However, common strategies to improve individual quality, such as fine-tuning for high quality, less diverse datasets (Rombach et al., 2022) or strong classifier-free guidance (CFG) (Ho & Salimans, 2021), often sacrifice diversity (Astolfi et al., 2024). Additional conditioning methods like spatial controls (Zhang et al., 2023) or image prompting (Ye et al., 2023) improve controllability but also reduce diversity, especially with strong guidance values. This lack of diversity is further worsened by distilled generators (Sauer et al., 2024a;b; Kang et al., 2024; Yin et al., 2024). Our work addresses this trade-off with group inference, enhancing both sample quality and diversity in batches, and demonstrating applicability across various controls (text, spatial, visual prompts). Generative Photomontage (Liu et al., 2025) generates diverse results by composing samples, but it requires additional user inputs.

**Diffusion Inference and Guidance.** Inference-time guidance improves quality and controllability of diffusion models without costly model finetuning. Early methods, such as classifier guidance (Dhariwal & Nichol, 2021) and widely-used classifier-free guidance (CFG) (Ho & Salimans, 2021), significantly increase sample quality, often at the cost of diversity. Recent approaches manipulate internal representations, such as cross-attention maps (Parmar et al., 2023), or incorporate spatial control from inputs such as layouts or sketches (Chen et al., 2024; Kim et al., 2023; Phung et al., 2024; Voynov et al., 2023; He et al., 2023). Other strategies apply guidance over limited intervals (Kynkäänniemi et al., 2024) or thresholding to CFG to reduce saturation (Saharia et al., 2022). Similarly, (Jena et al., 2025) studies the problem of reward hacking and how finetuning can lead to a loss of diversity.

While the above techniques improve individual samples, our group inference approach explicitly optimizes collective properties, balancing sample quality and inter-sample diversity (Cohen et al., 2024). A closely related work is particle guidance (Corso et al., 2023), which incorporates a pairwise potential during denoising steps to encourage diversity. Our method differs in three ways. First, we improve both quality and diversity, while particle guidance often hurts image quality (Section 4.2). Second, our method scales effectively to a large number of images through early candidate pruning and sample selection, avoiding expensive optimization. In contrast, particle guidance is limited to small sets (e.g., four images) due to memory-intensive pairwise gradient computation. Third, our framework supports non-differentiable scores, enabling metrics derived from multimodal LLMs.

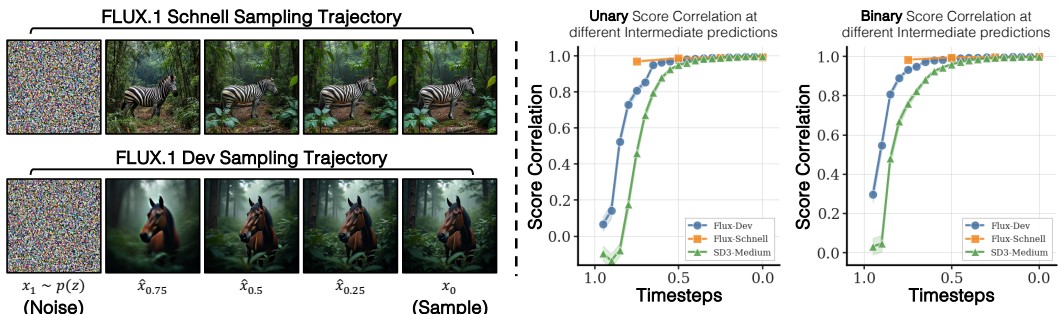

Figure 3: **Correlation Between Intermediate and Final Generation Scores.** On the left, we show the reverse diffusion process, visualizing the intermediate predictions $\hat{x}_t$ of the final image at different steps for FLUX.1 Schnell and FLUX.1 Dev models. Observe that intermediate predictions look similar to true final sample $\mathbf{x}_0$. We further demonstrate this quantitatively by plotting the Spearman correlation of the Unary and Binary scores from $\hat{x}_t$ versus final $\mathbf{x}_0$ scores, across different steps. For multistep models, the plots demonstrate strong correlations rapidly approaching 1.0, even at early timesteps. For distilled models (Flux-Schnell), the correlation is high from the first denoising step. This motivates using intermediate predictions for progressively pruning candidate samples.

Other related works such as CADS (Sadat et al., 2025), DiverseFlow (Morshed & Boddeti, 2025), Shielded Diffusion (Kirchhof et al., 2025), and NegToMe (Singh et al., 2024) also attempt to improve the diversity of generated samples, through negative guidance from a reference image. CADS approaches this by adding Gaussian noise to conditioning signal in a monotonically decreasing schedule. DiverseFlow and Shielded Diffusion respectively employ a determinantal point process, and group repellency to drive diversity of generated samples. All of these approaches modify the sampling trajectory, and as a result, produce samples that have lower quality.

**Inference-time Scaling.** Test-time scaling, leveraging methods like chain-of-thought (Wei et al., 2022), proposer and verifier (Snell et al., 2024), or multi-step reasoning, has become a key research area for large-language models (Muennighoff et al., 2025). Recently, researchers have adopted the inference-time scaling for diffusion models (Ma et al., 2025), which uses off-the-shelf models and evaluation metrics to search for better noises and increase the sample quality, often requiring thousands to tens of thousands of function evaluations (NFEs). However, text-to-image models differ from LLMs in three ways: they are often more computationally expensive (Li et al., 2024), users often pay 5 to 10 cents per image on leading platforms, and low latency is often required. Our method balances the inference computational cost with quality and diversity improvement.

## 3 METHOD

We propose *Scalable Group Inference*, a test-time selection framework that chooses a diverse, high-quality subset from a large pool of generated outputs. The method relies on a scoring objective that combines a unary term that measures the quality of an individual sample and a binary term that computes pairwise properties such as image distances. We first formulate this as a quadratic integer programming problem (QIP) over binary selection variables. Then, to reduce compute cost, we introduce a progressive filtering strategy that prunes low-quality candidates early using intermediate predictions from partially denoised samples. We now describe both components in detail.

### 3.1 FORMULATION

Given a generative model $G_\theta(\mathbf{z}, \mathbf{c})$ that maps latent noise $\mathbf{z} \sim p(\mathbf{z})$ and condition $\mathbf{c}$ to outputs $\mathbf{x}$, our goal is to obtain a *set* of $K$ outputs, $\{\mathbf{x}^{(i)}\}_{i=1}^K$, that exhibits both high quality and diversity together. We begin by generating a large set of $M$ candidate outputs $\{\mathbf{x}^{(i)}\}_{i=1}^M$ using i.i.d. sampling:

$$\mathbf{x}^{(i)} = G_\theta(\mathbf{z}^{(i)}, \mathbf{c}), \quad \mathbf{z}^{(i)} \overset{\text{i.i.d.}}{\sim} p(\mathbf{z}). \tag{1}$$

Let $\mathcal{I} = \{1, \dots, M\}$ index the candidate samples. We associate each sample $i \in \mathcal{I}$ with a unary score $\mathbf{u}_i \in \mathbb{R}$ and each pair $(i, j)$ with a binary score $\mathbf{b}_{ij} \in \mathbb{R}$. Concretely,

$$\mathbf{u}_i = f_{\text{CLIP}}(\mathbf{x}^{(i)}, \mathbf{c}) \tag{2}$$

$$\mathbf{b}_{ij} = 1 - \text{cosine}\left(f_{\text{DINO}}(\mathbf{x}^{(i)}), f_{\text{DINO}}(\mathbf{x}^{(j)})\right) \tag{3}$$

where $f_{\text{CLIP}}$ (Ramesh et al., 2022) computes the similarity between the input image and the target caption, and $f_{\text{DINO}}$ is the DINOv2 (Oquab et al., 2023) feature extractor. Note that our method is general and accommodate many different choices of score functions as discussed later in Section 4.4.

We introduce binary selection variables $\mathbf{y}_i \in \{0, 1\}$ where $\mathbf{y}_i = 1$ indicates that candidate $i$ is included in the next group. We define the group selection objective as:

$$\max_{\mathbf{y} \in \{0,1\}^M} \quad \sum_{i \in \mathcal{I}} \mathbf{u}_i \, \mathbf{y}_i + \lambda \sum_{\substack{i,j \in \mathcal{I} \\ i < j}} \mathbf{b}_{ij} \, \mathbf{y}_i \, \mathbf{y}_j$$

$$\text{subject to} \quad \sum_{i \in \mathcal{I}} \mathbf{y}_i = K. \tag{4}$$

$\lambda$ is the hyperparameter that controls the relative weight between the unary and binary scores. The first term rewards individually strong outputs; the second promotes diversity by favoring dissimilar pairs. Solving this QIP yields a subset of size $K$ with desirable group properties. We use the branch-and-cut algorithm implemented by an off-the-shelf solver (Gurobi Optimization, LLC, 2025) to solve the QIP. The formulation is model-agnostic and can accommodate any scoring functions, including functions that are not differentiable.

## 3.2 PROGRESSIVE PRUNING FOR EFFICIENT SELECTION

Naively applying group selection requires generating all $M$ candidates to completion, which is prohibitively expensive for recent large models like Flux Dev (Labs, 2024). For example, generating $M = 64$ samples over $T = 20$ denoising steps requires $M \cdot T$ forward passes. Even on a modern GPU like NVIDIA H100, this naive approach results in a runtime of more than 3 minutes. To reduce this cost, our progressive strategy that prunes candidates early using intermediate predictions.

**Intermediate pruning.** We maintain a set $\mathcal{S}_t \subseteq \mathcal{I}$ of candidate indices at each step $t$. For each sample in $\mathcal{S}_t$, we compute the intermediate prediction $\hat{x}_t$, evaluate the unary and binary scores, and solve the QIP (Eq. 4) to select the best subset. This subset becomes the next set $\mathcal{S}_{t-1}$, forming a nested sequence:

$$\mathcal{S}_T \supset \mathcal{S}_{T-1} \supset \cdots \supset \mathcal{S}_0.$$

Once the set reaches desired output group size $K$, we stop pruning and complete the remaining denoising steps only for selected samples. See Algorithm 1 in appendix for a detailed pseudo-code.

**Reliability of early predictions.** In modern multi-step diffusion and flow-based models, the intermediate state $\mathbf{x}_t$ already encodes coarse information about the final generated sample $\mathbf{x}_0$. A common approximation of the final image at timestep $t$ is the predicted reconstruction:

$$\hat{\mathbf{x}}_t = \mathbf{x}_t + t \cdot \epsilon_\theta(\mathbf{x}_t, t, \mathbf{c}), \tag{5}$$

where $\epsilon_\theta$ predicts the noise or velocity at time $t$. Although these predictions are coarse, they are sufficient for computing the unary and binary scoring functions.

To quantify this, we compute the correlation between the scoring functions (e.g., CLIP similarity or pairwise DINO diversity) evaluated on the intermediate images $\hat{\mathbf{x}}_t$ and the final output $\mathbf{x}_0$. Across a range of denoising steps, Figure 3 (right) shows strong correlations (e.g., $r > 0.7$ after 5 steps for multi-step models, and $r > 0.95$ after the first step for distilled models), indicating that intermediate predictions are reliable proxies. Figure 3 (left) shows this visually. Appendix Figure 17 shows a similar trend for other models. This high correlation enables us to safely rank candidates before they are fully denoised.

## 3.3 COMPUTATIONAL COMPLEXITY ANALYSIS

Suppose we start with $M$ candidates and prune the set by a fixed ratio $\rho \in (0, 1)$ at each denoising step until reaching a target set size $K$. The number of candidates at timestep $t$ is given by:

$$|\mathcal{S}_t| = \max \left( \rho^t M, K \right). \tag{6}$$

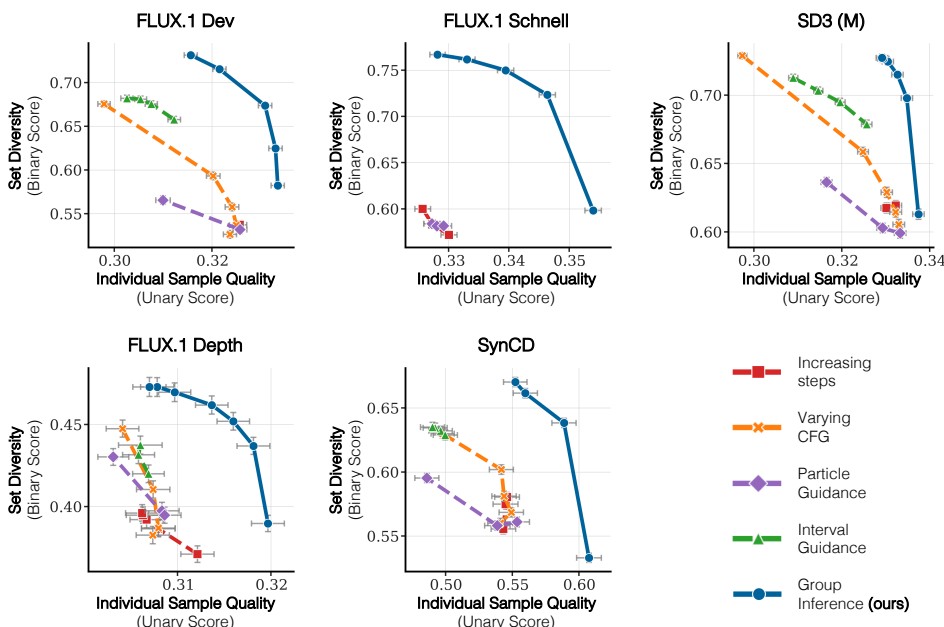

Figure 4: **Quality and Diversity Pareto front.** Each curve corresponds to a different inference strategy for five different models (FLUX.1 Dev, FLUX.1 Schnell, Stable Diffusion 3 Medium). Our proposed Group Inference (**blue**) consistently dominates alternate methods achieving Pareto optimality and superior tradeoffs between quality and diversity across all methods. Varying CFG and Interval Guidance do not apply to the distilled model (FLUX.1 Schnell).

Let $T$ denote the total number of denoising steps. We define the timestep $t^*$ at which the candidate set size first reaches or falls below the target $K$:

$$t^* = \left\lceil \frac{\log(K/M)}{\log(\rho)} \right\rceil. \tag{7}$$

The total number of model evaluations $f_\theta$ required throughout the process can be written as:

$$M \cdot \frac{1 - \rho^{t^*}}{1 - \rho} + K \cdot (T - t^* + 1). \tag{8}$$

In contrast, naive sampling without pruning would require $M \cdot T$ model evaluations. For typical parameter settings (e.g., $M = 64$, $K = 4$, $\rho = 0.5$, $T = 20$), our progressive filtering approach yields substantial compute savings (i.e., 184 vs 1280 evaluations, $\sim 85\%$ reduction). Our method has an overall complexity of $\mathcal{O}(M + KT)$.

## 4 EXPERIMENTS

We demonstrate the effectiveness of our scalable group inference across three different tasks: text-to-image generation, depth-conditioned generation, encoder-based image customization, and five different base models: FLUX.1 Schnell, FLUX.1 Dev, Stable Diffusion 3 (Medium), FLUX.1 Depth, and SynCD. Section 4.1 describes dataset and evaluation protocols used. Section 4.2 and Section 4.3 show a comparison with baselines. Appendix A.1, A.2, A.3 shows more results, analysis and ablations. Please see Appendix Figures 22, 19, 20, and 21 for additional visual results generated with our method. Note that our method does not cause an increase in peak-GPU memory usage even at early stages; it generates samples serially, one at a time.

### 4.1 DATASET AND EVALUATION

**Datasets.**   We use GenEval (Ghosh et al., 2023), validation split of COCO 2017 dataset (Lin et al., 2014), and DreamBooth dataset (Ruiz et al., 2023) for text-to-image generation, depth-conditioned generation, and image customization, respectively. For depth-conditioned generation, we first extract depth using a recent method Yang et al. (2024a).

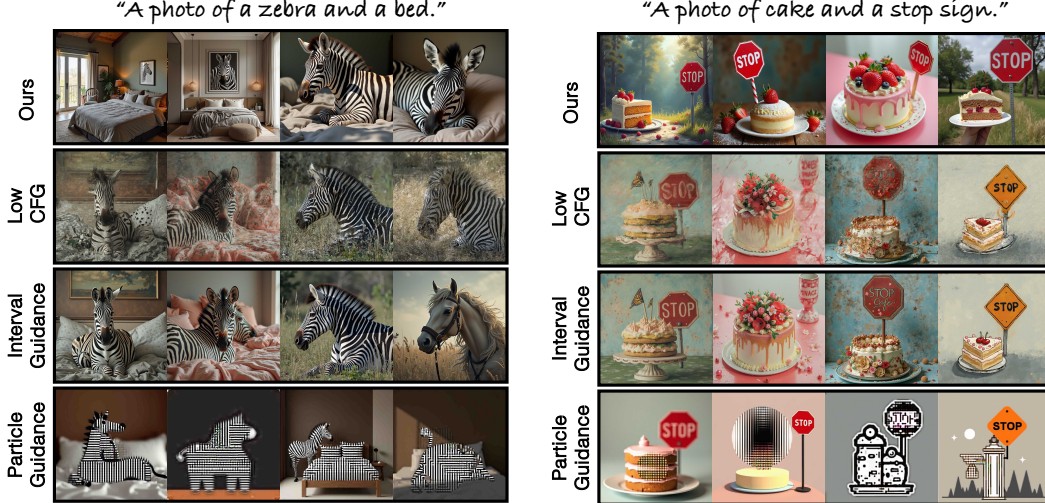

Figure 5: **Qualitative results.** We compare our method (top row) against baselines targeting an improved Quality-Diversity tradeoff with FLUX.1 Dev. For fair comparison, baselines were configured to approximate the diversity achieved by our approach. The result demonstrates that: (i) employing a low CFG scale to increase diversity results in diminished image quality; (ii) Interval Guidance exhibits reduced adherence to the input text prompt; and (iii) Particle Guidance, by actively altering sampling trajectories, tends to produce less natural images. In contrast, our method outputs a set of diverse outputs while maintaining good image quality and prompt fidelity.

| Model | Comparison | Diversity | | Quality | |
|---|---|---|---|---|---|
| | | Ours pref. | Baseline pref. | Ours pref. | Baseline pref. |
| FLUX.1 Dev | Ours vs Low-CFG | **88.3%** | 11.70% | **85.6%** | 14.4% |
| | Ours vs Interval Guidance | **53.4%** | 46.6% | **58.4%** | 41.6% |
| | Ours vs Particle Guidance | **81.2%** | 18.8% | **79.4%** | 20.6% |
| FLUX.1 Schnell | Ours vs Low-CFG | | N/A | | |
| | Ours vs Interval Guidance | | N/A | | |
| | Ours vs Particle Guidance | **55.5%** | 44.5% | **62.3%** | 37.7% |
| SD3 (M) | Ours vs Low-CFG | **76.8%** | 23.2% | **80.8%** | 19.20% |
| | Ours vs Interval Guidance | **58.1%** | 41.9% | **57.9%** | 42.10% |
| | Ours vs Particle Guidance | **78.9%** | 21.1% | **85.9%** | 14.1% |

Table 1: **User preference comparison.** User study results demonstrate that our method is consistently preferred over alternative inference strategies. Across three different text-to-image models (FLUX.1 Dev, FLUX.1 Schnell, and Stable Diffusion 3 Medium), users consistently chose our generations for both diversity and quality. Note that comparisons against Low-CFG and Interval Guidance are not applicable (N/A) for the distilled FLUX.1 Schnell model.

**Models.** We use several recent models, including FLUX.1 Dev (Labs, 2024) and Stable Diffusion 3 Medium (SD3-M) (Esser et al., 2024), which are flow-based models typically requiring 20-50 denoising steps. We also evaluate FLUX.1 Schnell, a timestep-distilled variant designed for efficient generation, typically using 1-8 steps. For depth-conditioned generation, we use FLUX.1 Depth, a model specifically trained for structural guidance based on depth maps. For customization, we use SynCD (Kumari et al., 2025), a recent encoder-based image prompting model.

**Score Functions.** We use CLIP (Radford et al., 2021) text-image similarity to assess the quality of the individual samples for text-to-image and depth-to-image generation. For encoder-based image customization, we use cosine DINOv2 (Oquab et al., 2023) similarity between the input subject image and the output generated images for the unary score. Diversity is computed for all tasks as one minus the cosine similarity between the DINOv2 patchwise features of all image pairs in the output set. We show evaluation with other metrics in Appendix Section A.1 and Figure 16. Our method can naturally accommodate a wide range of scores, fitting the user's needs (Section 4.4).

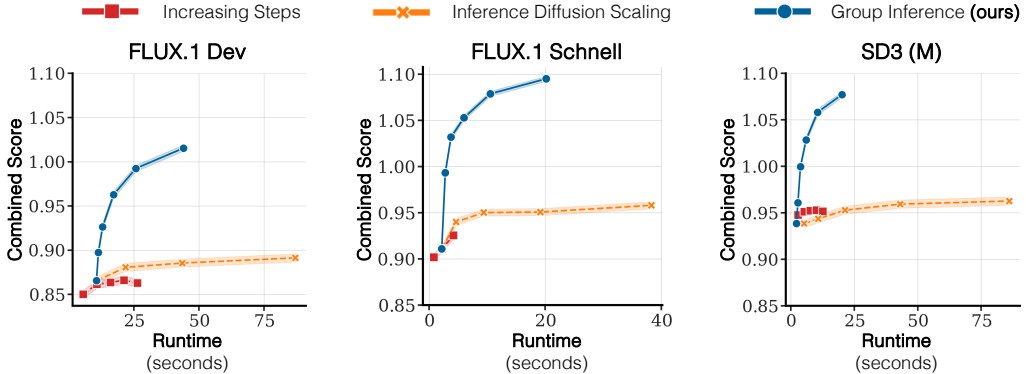

Figure 6: **Performance at different runtimes.** Increasing Steps (**red**) shows limited gains with additional computation. Inference Diffusion Scaling (Ma et al., 2025) (**orange**), which increases sample count through independent I.I.D. generations, requires substantially more runtime for marginal improvements. In contrast, our proposed Group Inference (**blue**) achieves significantly better performance–runtime tradeoffs, quickly outperforming both baselines with minimal overhead.

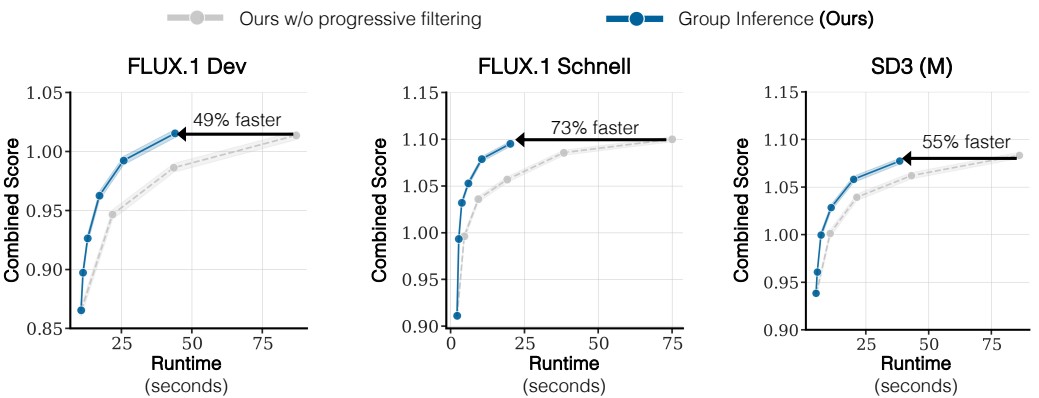

Figure 7: **Importance of progressive pruning.** In this ablation, we explicitly study the importance of progressive pruning. The gray line generates all candidate samples to completion, without pruning, and solved the QIP at the end to select 4 output samples. Across multiple base generative models, progressive pruning consistently enables our method to select candidates efficiently and shows substantial speedups-49%, 73%, and 55% faster for comparable combined group scores.

## 4.2 BASELINES AND THE DIVERSITY-QUALITY TRADEOFF

In generative modeling, tradeoff exists between optimizing for the perceptual quality of individual samples and ensuring a diverse set of outputs (Zhu et al., 2017; Ho & Salimans, 2021; Brock et al., 2018). Figure 4 plots the quality-diversity Pareto front, Table 1 shows pairwise user preference studies, and qualitative comparisons are shown in Figure 5. Across all baselines, our proposed method consistently achieves a superior diversity-quality tradeoff. As illustrated by the **blue** line in Figures 4, our approach dominates the Pareto fronts, yielding better diversity for a given quality, or higher quality for a comparable diversity. Appendix Figures 16, 23 show additional comparisons. We have included additional comparisons with CADS (Sadat et al., 2025), Shielded Diffusion (Kirchhof et al., 2025), DiverseFlow (Morshed & Boddeti, 2025), and NegToMe (Singh et al., 2024) in the appendix.

**Increasing Denoising Steps.** We first consider the impact of simply increasing the denoising steps. While more steps can sometimes refine details, this has a minimal effect on meaningfully shifting the diversity-quality balance, as shown by the **red** line in Figures 4.

**Varying CFG.** Next, we examine the widely used technique of varying the Classifier-Free Guidance scale (Ho & Salimans, 2021) (CFG). As depicted by the **orange** line in Figures 4, this traces a distinct tradeoff curve. Notably, low CFG values (e.g., CFG=1) largely increase output diversity but often at the cost of a sharp degradation in sample quality and prompt alignment. This is visually

seen through the poor image quality in the second row in Figure 5, where a low CFG value is used. Table 1 user study further corroborate these observations.

**Interval Guidance.** Interval guidance (Kynkäänniemi et al., 2024) applies CFG only to a subset of the timesteps. We conduct a sweep across various interval configurations (**green** line). Consistent with the third row in Figure 5, this approach can offer image quality improvements over a standard CFG sweep. However, it still performs worse than our method in terms of both quality and diversity. In the example of zebra and bed on the left in Figure 5, Interval Guidance reduces diversity of zebra poses and does not generate a bed for two of the four outputs. Similarly, in the right example, interval guidance does not always generate a stop sign. Moreover, both CFG sweeping and Interval Guidance do not apply to distilled models as they do not use guidance mechanisms.

**Particle Guidance.** Particle Guidance (Corso et al., 2023) optimizes a binary potential to encourage diversity. Following the original work, we use DINO features for the diversity term. As illustrated by the **purple** line, Particle Guidance can indeed increase output diversity. However, this comes with a sharp decrease in individual sample quality (fourth row in Figure 5), as direct optimization of the binary potential actively alters the sampling trajectory. This can push the output samples off the learned data manifold, leading to less natural and artifact-prone images. Furthermore, Particle Guidance incurs a substantial memory cost due to the necessity of computing gradients and backpropagating through the binary potential. This reliance on gradient computation also makes the method unsuitable for non-differentiable potential functions.

## 4.3 INFERENCE SCALING ANALYSIS

Scaling computational resources at the test time to enhance model performance is an increasingly useful paradigm in machine learning. For diffusion models, a native mechanism for test-time scaling involves increasing the number of denoising steps. Although this can initially lead to improved sample quality, this approach often yields diminishing returns; beyond a certain point, additional denoising steps provide progressively smaller gains in quality. In Figure 6, we illustrate the impact of various test-time scaling strategies on the group objective (Equation 4).

Our first baseline (Figure 6, **red** line) increases the denoising steps for a fixed initial samples $M$. Consistent with existing findings (Ma et al., 2025), this approach demonstrates minimal improvement in our combined score, with the curve quickly plateauing. Inference Diffusion Scaling (Ma et al., 2025), uses the additional compute budget to perform a search over multiple random seeds. For a fair comparison, we implement this baseline with the CLIP text-image similarity as the verifier. This method does not incorporate intermediate predictions for their final text-to-image results and does not consider any pairwise terms. Consequently, it is not effective in improving the group objective, as shown through the **orange** line. In contrast, our method invests scales by increasing the number of initial samples and produces consistent improvements as depicted by the **blue** line in Figure 6.

**Ablating progressive filtering.** In Figure 7, we compare our complete method, which utilizes progressive filtering with intermediate predictions $\hat{x}_t$ (**red** line), against a variant that performs full denoising for all $M$ candidate samples without such filtering (**gray** line). Demonstrating its effectiveness across different architectures, our approach achieved comparable group scores while requiring up to 73% less runtime.

## 4.4 DIFFERENT DIVERSITY OBJECTIVES

Next, we show that our approach is general and can accommodate different pairwise binary objectives by simply swapping the binary term in our quadratic integer programming objective, as demonstrated in Figure 8. Let us consider the example shown on the left corresponding to the caption "a giant neon rose." Standard I.I.D. sampling (top row) produces a set of visually redundant images. All four roses are red and share a similar pose. In contrast, the bottom two rows are generated using our method with an identical unary quality term (CLIP text-image similarity) but different binary diversity objectives. The middle row uses a direct color-based dissimilarity as the binary term. This successfully steers the outputs towards a set of images with varied and distinct color schemes. For the rose example on the left, this results in a vibrant set that includes blue, orange, and pink neon variants. In the bottom row, we use a DINO diversity metric that captures more semantic features when comparing the pairwise distances. This change directs the model to produce a set with higher structural variance. As seen with the rose example, this yields outputs with different poses and camera angles. This direct comparison underscores a key strength of our approach: the ability to seamlessly integrate different notions of diversity to achieve targeted, user-defined visual outcomes.

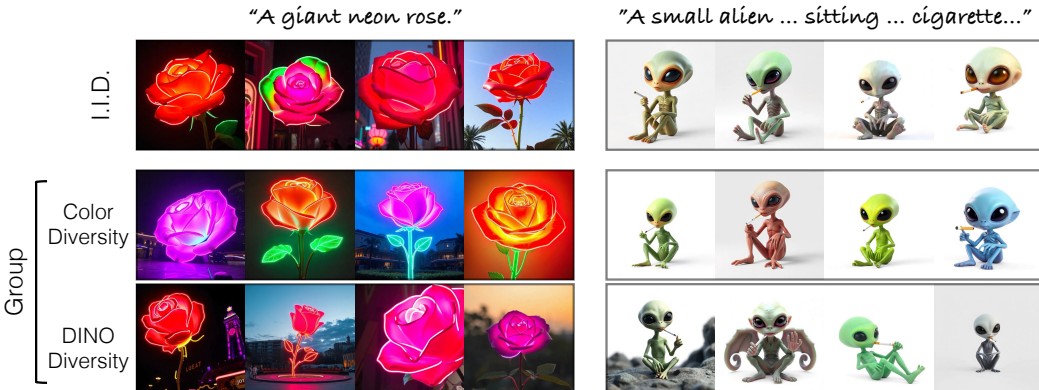

Figure 8: **Different pairwise objectives.** Compared to I.I.D. (top row), our method allows for targeted diversity by defining different pairwise objectives. Unary quality term is identical but pairwise binary term is varied in second and third rows. The middle row uses a color-based binary term, while bottom row uses a DINO-based binary term.

## 5 CONCLUSION

In this paper, we have introduced scalable group inference, a novel method to generate diverse, high-quality sets of samples by formulating the selection as a quadratic integer program and leveraging intermediate predictions for improving the runtime efficiency. Our efficient approach significantly enhances group diversity and quality compared to existing baselines across various generative tasks. Still, our method has several limitations such as reliance on the base model and the score functions being efficient. We expand on our limitations in Appendix Section A.4 and Figure 15.

## 6 ETHICS STATEMENT

Our work introduces a scalable group inference method designed to enhance the diversity and quality of outputs from generative models. We recognize that generative models can be used to produce harmful, biased, or misleading content. However, our method operates at the inference stage and does not alter the underlying model, its training data, or its inherent biases. The primary goal of our work is to improve the user experience by providing a more varied and high-quality set of candidate outputs for a given prompt, thereby fostering greater creative exploration. By discouraging redundant outputs, our approach encourages a more thoughtful and efficient use of generative systems.

## 7 REPRODUCIBILITY OF RESULTS

We provide all details of the experiments done in our paper, including the datasets used, model and baseline inference parameters, score computation, in Section 3 and appendix Section A.3. We have also provided a full pseudocode for our method in appendix Algorithm 1. An anonymized version of the code is also included as a zipped file in the accompanying supplementary materials. These details and resources should allow other researchers to reproduce all results we have shown in this paper.

**Acknowledgment.** We thank Daniel Cohen-Or and Sheng-Yu Wang for their helpful comments and discussion. We are also grateful to Nupur Kumari for proofreading the draft. The project is partially supported by Snap Research, NSF IIS-2239076, DARPA ECOLE, and the Packard Fellowship.

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

# A APPENDIX

Section A.2 presents additional qualitative and quantitative results obtained by our method across multiple different base generative models and tasks. Section A.1 provides an additional analysis of the different components of our method. Section A.3 details the datasets and implementation settings used for each of the baseline methods. Section A.4 discusses the broader impacts and limitations of our method.

## A.1 ANALYSIS

In this section, we provide additional analysis of the different components of our method.

**Runtime breakdown.** In Figure 9 we show the runtime breakdown of different steps in the pipeline for the FLUX.1 Dev base model using CLIP text image similarity as the unary score and DINOv2 diversity as the binary score. On the left, we fix the output set size K to be 4 and increase the initial candidate size from 4 to 200. On the right, we fix the initial candidate size to be 200, and increase the output set size from 4 to 128. Note that across all settings, the runtime cost incurred by the QIP solver and the score computation is negligible compared to the forward pass of the denoising transformer. Our method denoises one candidate at a time and does not incur any additional inference memory costs even at early timesteps when the candidate size is large.

**Different $\rho$ values.** Figure 12 illustrates the effects of varying pruning ratios ($\rho$) on the FLUX.1 Dev and FLUX.1 Schnell models. The figure presents both the Number of Function Evaluations (NFE) (left plot) and the wallclock runtime on a single NVIDIA H100 (right plot). Across all plots, a pruning ratio of $\rho = 1.0$ signifies no progressive pruning. For the FLUX.1 Dev model, lower pruning ratios (e.g., $\rho = 0.1$ and $\rho = 0.25$) are overly aggressive, leading to suboptimal scores. Conversely, a pruning ratio of $\rho = 1.0$ (no candidate filtering) achieves a good combined score but incurs a high inference cost. A pruning ratio of $\rho = 0.5$ strikes an effective balance, yielding higher scores without excessive computational cost. We use $\rho = 0.5$ for all FLUX.1 Dev experiments.

A different trend observed for distilled FLUX.1 Schnell model. It can accommodate a more aggressive pruning ratio, such as $\rho = 0.1$, without a noticeable decrease in the score. This can be attributed to the better reliability of the intermediate predictions for the distilled models, as shown in Figure 3 of the main paper.

**Efficient decoding.** Our method uses an efficient decoder Bohan (2023) to decode all intermediate predictions for progressive pruning. In Figure 14 we ablate the use of efficient decoder and show that across both FLUX.1 Dev and FLUX.1 Schnell, using an efficient decoder improves the runtime without sacrificing the score.

**Evaluation with different score functions.** Figure 4 in the main paper shows the quality and diversity Pareto front for the text to image generation task. That figure uses CLIP text-image similarity (Equation 2) as the quality score and DINO diversity 3 as the diversity score. Next, we evaluate our method using several additional score functions that are not used by our method for selection in Figure 16. The top row uses Image Reward Xu et al. (2023) for measuring quality of samples and depth features to measure diversity. Image Reward is a network that is trained to learn human preferences for text-to-image generation. The depth diversity is calculated with the DepthAnything V2 model Yang et al. (2024b). The bottom row uses BLIP2 Li et al. (2023) to measure the quality and CLIP features to compute the diversity. Figure 16 shows a comparison with three different base models: FLUX.1 Dev (left), FLUX.1 Schnell (middle), and Stable Diffusion 3 medium (right). Across each model, our proposed group inference shows a better trade off between quality and diversity. Note that particle guidance obtains slightly better BLIP2 score than our method for Stable Diffusion 3 (M). However, the outputs generated by particle guidance have artifacts. This is also reflected by a low score for other metrics (Image Reward and CLIP), and a worse user preference score.

## A.2 ADDITIONAL RESULTS

**Qualitative results.** In Figures 19, 20, and 21 we show additional visual examples of our method. Across multiple models and tasks, our method consistently outputs samples that are more diverse, and without any degradation in the quality. Similar to the Figure 5 in the main paper, Figure 23 shows additional visual comparison to baselines. In these figures, the Low CFG baseline uses a

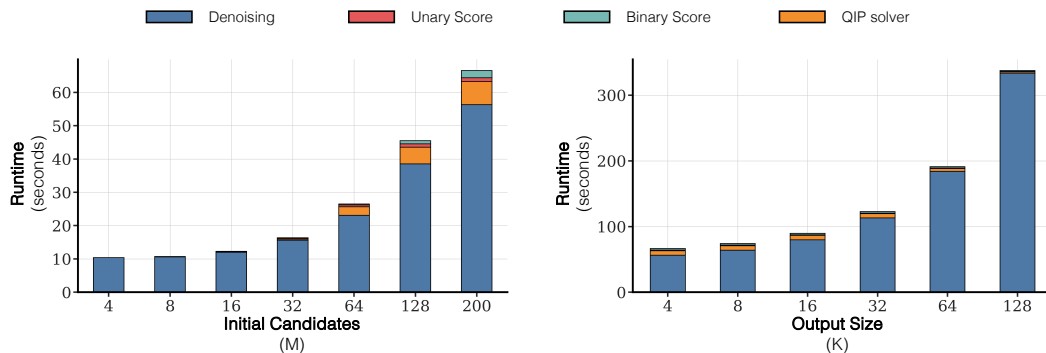

Figure 9: **Runtime breakdown.** We show a runtime breakdown of our method using FLUX.1 Dev model as the number of initial candidates (M, left) and the output set size (K, right) is increased. On the left plot, the output size is fixed to 4 and in the right plot the initial candidate size is fixed to 200. Across all settings, the runtime is dominated by the denoising step.

CFG value of 1.0. Interval Guidance uses an interval of $[0.6, 0.4]$, and particle guidance uses a coefficient value of 100.

**Correlation analysis.** Figure 3 in the main paper shows the correlation between the scores computed with the final image and the intermediate images. In Figure 17, we show that a similar correlation trend is visible in other base models (FLUX.1 Depth and SynCD). FLUX.1 Depth and SynCD show the CLIP text image similarity as the unary scores, and DINO diversity as the binary scores. This is consistent with our observations in the main paper.

## A.3 IMPLEMENTATION DETAILS

Section A.3.1 first provides implementation details and hyperparameters used for all settings shown in the main paper. Section A.3.2 lists details about the datasets used for each task.

### A.3.1 BASELINES
**Increasing steps.** For FLUX.1 Dev, Stable Diffusion 3 Medium, FLUX.1 Depth, and SynCD, we consider the timesteps 10, 20, 30, 40, and 50. For the distilled model, FLUX.1 Schnell, we consider the timesteps 1, 2, 4, and 8.

**Varying CFG.** For FLUX.1 Dev, FLUX.1 Depth, and SynCD we consider the CFG values 1, 2, 3, 4, and 5. For Stable Diffusion 3 Medium, we consider the CFG values 1, 5, 10, and 15. Note that FLUX.1 Schnell does not use CFG.

**Interval guidance.** For FLUX.1 Dev, FLUX.1 Depth, Stable Diffusion 3 Medium, and SynCD we consider the guidance intervals $[0.9, 0.1]$, $[0.8, 0.2]$, $[0.7, 0.3]$, and $[0.6, 0.4]$. Note that FLUX.1 Schnell does not use CFG, and this baseline is not applicable.

**Particle guidance.** For the Particle Guidance baseline, we consider coefficient values 0, 10, 50, 100, and 200. Note that this baseline significantly increases the memory consumption during inference.

**CADS Sadat et al. (2025).** We follow the implementation details and hyperparameters provided by authors. We use $\tau_1 = 0.6$, $\tau_2 = 0.9$, and $s = 0.25$. In Figure 10, we consider several $\psi$ values $(0, 0.5, 1.0)$.

**DiversityFlow Morshed & Boddeti (2025).** We implement this baseline following the details provided in the paper. Figure 10 considers different scale values 10, 100, 200.

**ShieldedDiffusion Kirchhof et al. (2025).** We implement Shielded Diffusion following the algorithm and hyperparameters provided in the appendix of the paper. We set the overcompensation value to be 1.6 and vary the radius values to be 100, 200, 500. Figure 10 plots the different radius values.

**NegToMe Singh et al. (2024).** We use the official released implementation for NegToMe.

---

**Algorithm 1** Efficient group inference

---

```
# model: The diffusion model ε_θ
# zs: Initial noise vectors {z_i}
# N: Total number of denoising steps
# ts: The noise schedule {t_j}
# K: The target number of samples
# c: Conditioning information
# rho: The dropping ratio ρ for pruning
def group_inference(model, zs, N, ts, K, c, rho):
    # Initialize the set of candidates from noise
    candidate_set = list(zs)

    # Denoising loop
    for j in reversed(range(1, N +1)):
        intermediate_previews, next_latents = [], []

        # Get intermediate previews for all candidates
        for x_t in candidate_set:
            preview, x_next = denoise(x_t, ts[j], ts[j-1], c, model)
            previews.append(preview)
            next_latents.append(x_next)

        if len(candidate_set) > K:
            # Score previews and select the best subset
            u = unary_score(previews)
            b = binary_score(previews)

            # Prune candidates based on the dropping ratio rho
            m = max(K, int(len(candidate_set) * (1 - rho)))

            indices = SolveQIP(u, b, m)
            candidate_set = [next_latents[i] for i in indices]
        else:
            candidate_set = next_latents

    return candidate_set
```

---

**Inference diffusion scaling Ma et al. (2025).**    Figure 6 of the main paper shows a comparison to Inference Diffusion Scaling Ma et al. (2025), a concurrent work, that shows an improvement in the quality of samples. We follow the results in their paper and use random search as the strategy. For a fair comparison, we use the same CLIP text-image-similarity as the verifier.

**Group inference (ours).**    In Figures 4, and 16 of the main paper, we vary the $\lambda$ defined in Equation 4 while keeping the input samples $M$ fixed. We use $M = 128$ for FLUX.1 Dev, FLUX.1 Schnell, SD3 (M), and SynCD. Note that we additionally disable the classifier free guidance for the first denoising step for multi-step methods such as FLUX.1 Dev and FLUX.1 Depth. Note that varying the weighting factor $\lambda$ does not change the runtime, and only shows the trade-off between the diversity of samples in the generated output set and the individual quality. For FLUX.1 Dev, FLUX.1 Depth, SynCD we use $\rho = 0.5$. For SD3 (M) we use a higher $\rho = 0.75$, and for timestep distilled model FLUX.1 Schnell $\rho = 0.1$ in all experiments.

In Figures 6, 11 and 7 of the main paper, we want to study the performance at different runtimes, and therefore we fix the weighting factor $\lambda = 1$ but vary the number of input samples $M$ from 4 to 128.

**Choice of scores.**    Unless specified otherwise, FLUX.1 Dev, FLUX.1 Schnell, SD3 (M), Flux.1 Depth use CLIP text-image similarity as the unary score, and DINO diversity as the binary score. SynCD uses DINO target image similarity as the unary score, and DINO diversity as the binary score across all results. Figure 8 keeps the unary score fixed as the CLIP text-image similarity and shows the effects of varying the binary score function.

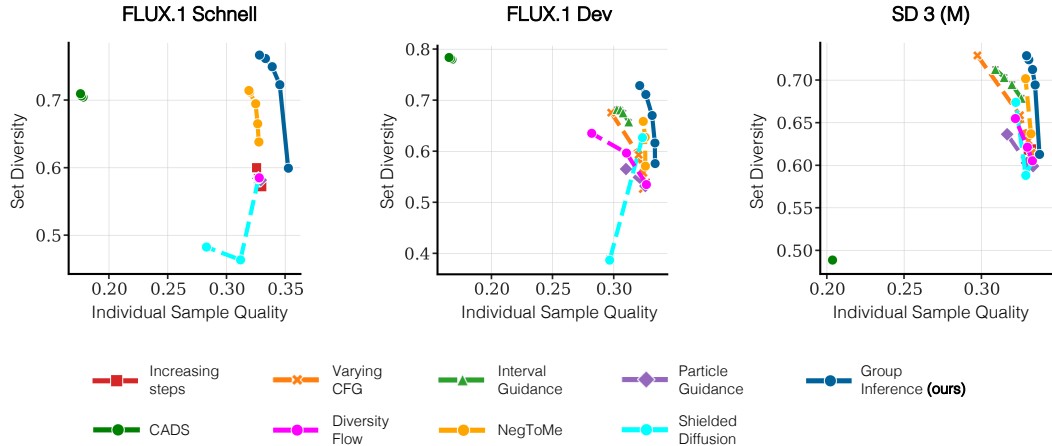

Figure 10: **Quality and Diversity Pareto front.** Each curve corresponds to a different inference strategy for three different models (FLUX.1 Dev, FLUX.1 Schnell, Stable Diffusion 3 Medium). Our proposed Group Inference (**blue**) consistently dominates alternate methods achieving Pareto optimality and superior tradeoffs between quality and diversity across all methods. Varying CFG and Interval Guidance do not apply to the distilled model (FLUX.1 Schnell).

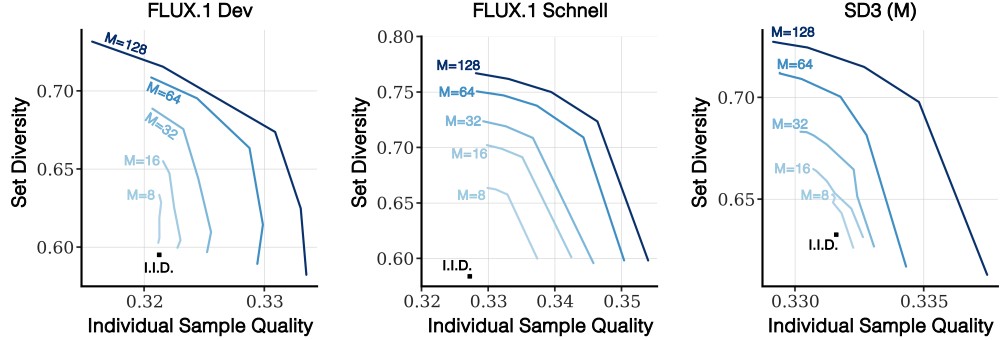

Figure 11: **Improvements as the number of initial samples M is increased.** We show how the sample quality (measured with CLIP) and the set diversity (measured with DINO) improves as the number of initial starting samples is increased from 4 to 128.

**Runtime.** We measure inference runtime using wallclock time. Specifically, this is the time taken by each method to generate an output set of $K$ images (where $K = 4$, unless specified otherwise) from a given input condition (i.e., a text prompt, depth map, or subject image). This measurement excludes initial model loading times and is averaged over 20 independent runs for each reported value. All runtime experiments utilize a single NVIDIA H100 GPU.

**Uncertainty Estimation.** We report standard errors for all quantitative results presented throughout our experiments. These standard errors are computed via bootstrapping with 1000 resamples.

**User Study.** We conduct two user preference studies to compare our method against each baseline on text-to-image generation. The first user study evaluates output diversity. In each comparison, the users are presented with two sets of 4 output images generated by two methods. The users are instructed to choose the set that has the higher variety. The second user study evaluates individual sample quality. For this study, the users are shown two images generated by two methods and asked to pick the one with higher quality. Both studies use Amazon Mechanical Turk (AMT). Each comparison was rated by three unique users, resulting in a total of 23,226 preference judgments.

**Additional Metrics.** Tables 2, 3, and 4 compare our approach with prior methods along additional image quality (HPSv2 (Wu et al., 2023), PickScore (Kirstain et al., 2023), ImageReward (Xu et al., 2023), and FID (Heusel et al., 2017), (Parmar et al., 2022)) and diversity metrics (Vendi score (Friedman & Dieng, 2022) and MSS). We use the COCO validation dataset for all FID results.

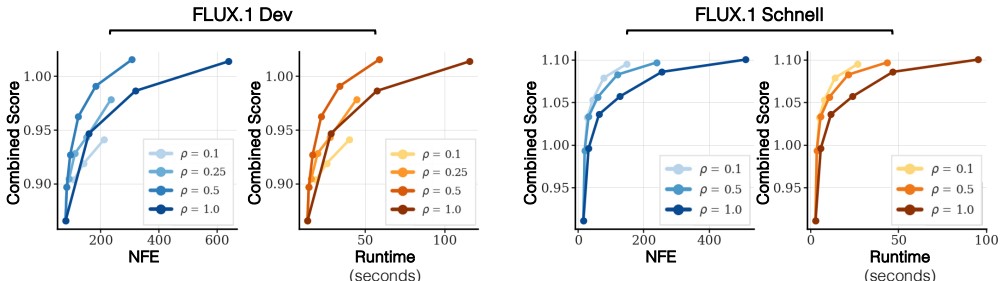

Figure 12: **Effects of different dropping ratio $\rho$.** We show the effects of different dropping ratios $\rho$ for two different base models: FLUX.1 Dev and FLUX.1 Schnell.

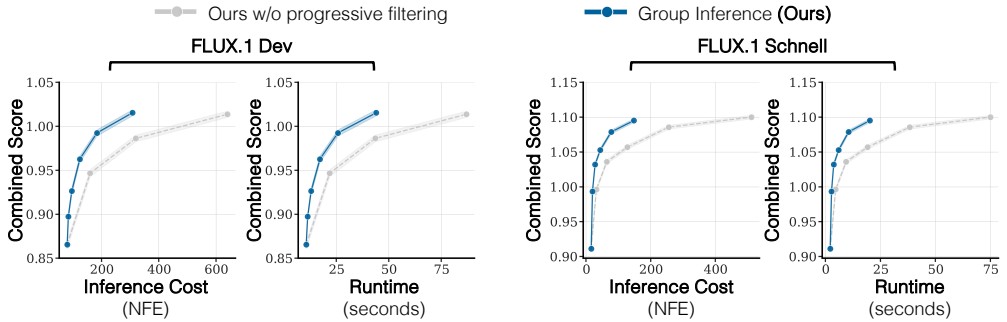

Figure 13: **Ablating the effect of progressive pruning.** Similar for Figure 7 from the main paper, we show the importance of progressive pruning. We report both, the number of function evaluations (NFEs) and the wallclock runtime (using one NVIDIA H100). The two plots on the left show the comparison using FLUX.1 Dev. The two plots on the right show FLUX.1 Schnell comparison.

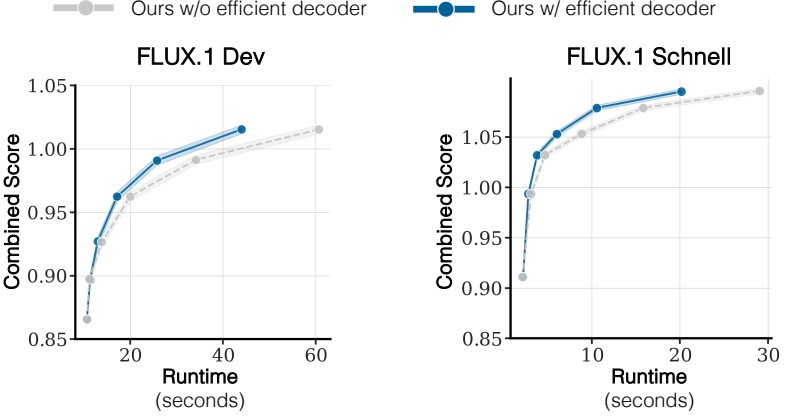

Figure 14: **Ablating efficient decoder.** We show the effects of using an efficient decoder for decoding the intermediate predictions.

### A.3.2 DATASET

**Text to image generation.** All text-to-image generation results with models FLUX.1 Dev, FLUX.1 Schnell, and SD3 (M) use all 553 prompts from the GenEval dataset Ghosh et al. (2023). Note that we use 250 COCO validation prompts for computing FID.

**Depth to image generation.** All FLUX.1 Depth experiments use depth maps computed using Depth Anything Large Yang et al. (2024a) from 250 images from the validation split of the COCO 2017 dataset Lin et al. (2014).

**Encoder-based image customization.** For all encoder-based image customization experiments using SynCD Kumari et al. (2025), we use 400 samples from the images in the standard DreamBooth dataset.

| Method | Quality | | | | | Diversity | | |
|---|---|---|---|---|---|---|---|---|
| | CLIP ↑ | HPSv2 ↑ | PickScore ↑ | Image Reward ↑ | FID ↓ | DINO ↑ | Vendi ↑ | MSS ↓ |
| I.I.D. | 0.327 | **0.302** | **0.235** | 1.028 | 21.22 | 0.584 | 3.173 | 0.562 |
| Particle Guidance | 0.329 | 0.301 | 0.234 | 1.043 | 19.55 | 0.581 | 3.166 | 0.564 |
| CADS | 0.176 | 0.124 | 0.179 | -2.224 | 51.00 | 0.710 | 3.522 | 0.468 |
| NegToMe | 0.319 | 0.272 | 0.224 | 0.608 | 21.00 | 0.714 | 3.567 | 0.464 |
| Shielded Diffusion | 0.327 | 0.302 | 0.235 | 1.028 | 21.26 | 0.584 | 3.174 | 0.562 |
| Diversity Flow | 0.328 | 0.302 | 0.234 | 1.030 | 20.48 | 0.585 | 3.178 | 0.561 |
| Ours | **0.341** | 0.298 | 0.234 | **1.071** | **18.76** | **0.744** | **3.644** | **0.442** |

Table 2: **Quality and Diversity Comparison with FLUX.1 Schnell.** We compare our method to several baseline sampling methods for the generation quality and group diversity using the FLUX.1 Schnell as the base model. Across all baselines, only our method is able to substantially improve the group diversity without sacrificing the quality of the samples. For instance, the vanilla I.I.D. inference has good sample quality but substantially worse diversity. Please refer to Figure 10 for a visual illustration of the pareto frontier. Note that the FID results are computed using the COCO validation dataset.

| Method | Quality | | | | | Diversity | | |
|---|---|---|---|---|---|---|---|---|
| | CLIP ↑ | HPSv2 ↑ | PickScore ↑ | Image Reward ↑ | FID ↓ | DINO ↑ | Vendi ↑ | MSS ↓ |
| I.I.D. | 0.324 | 0.303 | 0.237 | 0.953 | 23.77 | 0.558 | 3.077 | 0.582 |
| Interval Guidance | 0.305 | 0.270 | 0.226 | 0.294 | **19.12** | 0.681 | 3.462 | 0.489 |
| Particle Guidance | 0.276 | 0.199 | 0.204 | -0.699 | 58.28 | 0.564 | 3.101 | 0.577 |
| CADS | 0.165 | 0.139 | 0.177 | -2.204 | 51.91 | **0.784** | **3.714** | **0.412** |
| NegToMe | 0.324 | 0.301 | 0.234 | 0.915 | 23.02 | 0.663 | 3.412 | 0.503 |
| Shielded Diffusion | 0.323 | 0.294 | 0.231 | 0.907 | 20.02 | 0.627 | 3.313 | 0.530 |
| Diversity Flow | 0.282 | 0.214 | 0.208 | -0.432 | 40.04 | 0.635 | 3.301 | 0.524 |
| Ours | **0.326** | **0.307** | **0.237** | **1.001** | 23.21 | 0.711 | 3.552 | 0.466 |

Table 3: **Quality and Diversity Comparison with FLUX.1 Dev.** We compare our method to several baseline sampling methods for the generation quality and group diversity using the FLUX.1 Dev as the base model. Across all baselines, only our method is able to substantially improve the group diversity without sacrificing the quality of the samples. For instance, CADS obtains a high diversity score but has significantly degraded sample quality. Please refer to Figure 10 for a visual illustration of the pareto frontier. Note that the FID results are computed using the COCO validation dataset.

| Method | Quality | | | | | Diversity | | |
|---|---|---|---|---|---|---|---|---|
| | CLIP ↑ | HPSv2 ↑ | PickScore ↑ | Image Reward ↑ | FID ↓ | DINO ↑ | Vendi ↑ | MSS ↑ |
| I.I.D. | **0.333** | 0.288 | **0.232** | **0.973** | 19.48 | 0.605 | 3.233 | 0.546 |
| Interval Guidance | 0.315 | 0.241 | 0.220 | 0.121 | 19.49 | 0.703 | 3.518 | 0.473 |
| Particle Guidance | 0.317 | 0.257 | 0.216 | 0.583 | 22.85 | 0.636 | 3.518 | 0.523 |
| CADS | 0.202 | 0.091 | 0.179 | -2.270 | 86.75 | 0.491 | 2.799 | 0.632 |
| NegToMe | 0.328 | 0.269 | 0.224 | 0.678 | **18.19** | 0.706 | 3.534 | 0.471 |
| Shielded Diffusion | 0.329 | 0.280 | 0.229 | 0.803 | 20.35 | 0.588 | 3.144 | 0.559 |
| Diversity Flow | 0.322 | 0.272 | 0.223 | 0.719 | 18.79 | 0.655 | 3.385 | 0.509 |
| Ours | 0.333 | **0.288** | 0.231 | 0.930 | 18.34 | **0.712** | **3.558** | **0.466** |

Table 4: **Quality and Diversity Comparison with SD3 (M).** We compare our method to several baseline sampling methods for the generation quality and group diversity using the Stable Diffusion 3 (M) as the base model. Across all baselines, only our method is able to substantially improve the group diversity without sacrificing the quality of the samples. For instance, the vanilla I.I.D. inference has good sample quality but substantially worse diversity. Please refer to Figure 10 for a visual illustration of the pareto frontier. Note that the FID results are computed using the COCO validation dataset.

## A.4 BROADER IMPACTS, AND LIMITATIONS

Our work has a few limitations. First, our method relies on the base generative model's ability to produce a sufficiently diverse and high-quality initial candidate pool. Consequently, if the underlying model generates outputs of inherently poor quality or suffers from significant mode collapse, the efficacy of Scalable Group Inference in identifying an optimal set will be inherently constrained, as

| Unary Score Used | IID | Group Inference |
|---|---|---|
| Image Reward | 1.028 | **1.451** |
| HPSv2 | 0.302 | **0.315** |

Table 5: **Robustness to Unary Score with FLUX.1 Schnell.** We evaluate how different unary scoring functions affect IID and Group Inference performance when using FLUX.1 Schnell as the base model. The results show that our group inference method improves quality for both ImageReward (Xu et al., 2023) and HPSv2 Wu et al. (2023) unary scores.

| Unary Score Used | IID | Group Inference |
|---|---|---|
| Image Reward | 0.953 | **1.303** |
| HPSv2 | 0.303 | **0.311** |

Table 6: **Robustness to Unary Score with FLUX.1 Dev.** We evaluate how different unary scoring functions affect IID and Group Inference performance when using FLUX.1 Dev as the base model. The results show that our group inference method improves quality for both ImageReward (Xu et al., 2023) and HPSv2 Wu et al. (2023) unary scores.

| Unary Score Used | IID | Group Inference |
|---|---|---|
| Image Reward | 0.973 | **1.126** |
| HPSv2 | 0.288 | **0.293** |

Table 7: **Robustness to Unary Score with SD-3 M.** We evaluate how different unary scoring functions affect IID and Group Inference performance when using Stable Diffusion 3 (M) as the base model. The results show that our group inference method improves quality for both ImageReward (Xu et al., 2023) and HPSv2 Wu et al. (2023) unary scores.

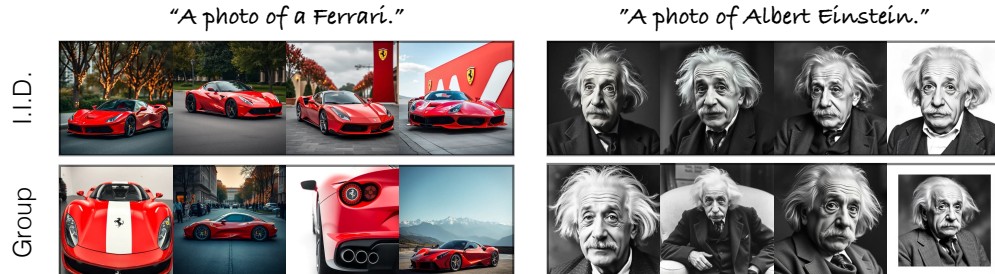

Figure 15: **Failure cases.** The performance of our method depends on the diversity of the initial candidate pool. (Left) For the prompt "A photo of a Ferrari," the base model (FLUX.1 Schnell) exhibits a strong color bias, exclusively generating red cars. Consequently, our method can find varied poses but is unable to produce a color-diverse set. (Right) Similarly, for "a photo of Albert Einstein," the base model only generates black-and-white images, constraining our method from finding any color photographs.

our method selects from, rather than intrinsically enhances, these initial candidates. This is visually illustrated in Figure 15.

Second, our method assumes that the unary (quality) and binary (diversity) scores are fast to compute. If evaluating these scores, especially the pairwise diversity metric across a large candidate set, is computationally intensive, the runtime benefits of our scalable optimization would be reduced.

Nevertheless, our method offers a path to more user-centric systems that efficiently output diverse, high-quality sets of options, and enhance creative exploration. This capability can significantly reduce the iterative burden in content generation across various domains. Concurrently, the increased efficiency in generating diverse sets of synthetic media could also have potential for misuse, such

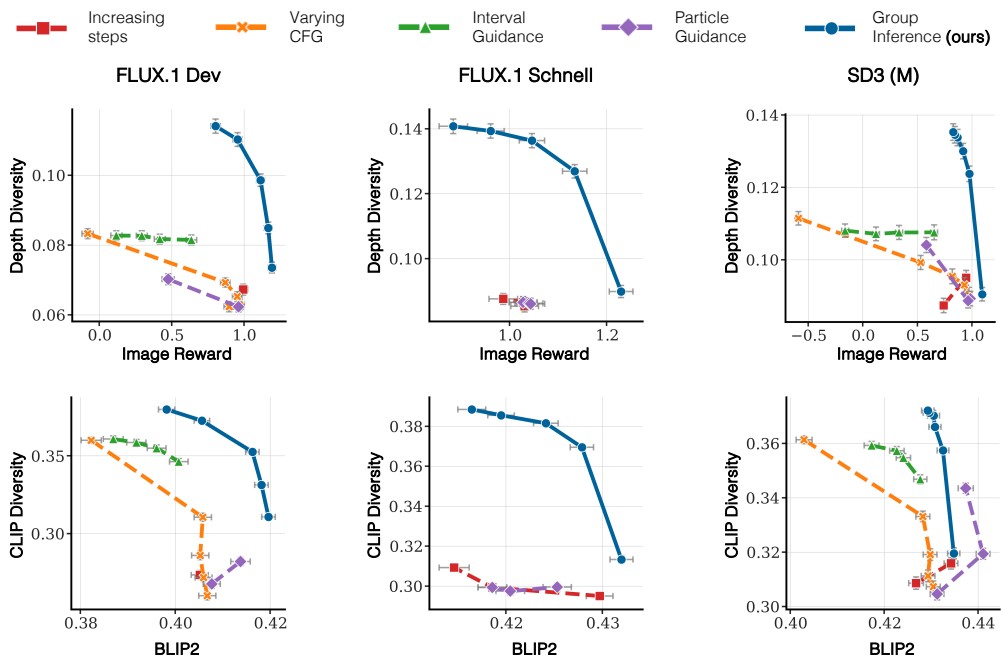

Figure 16: **Quality and Diversity Pareto front with additional metrics.** We evaluate the quality and diversity of samples generated by different inference strategies for three text-to-image models (FLUX.1 Dev, FLUX.1 Schnell, and Stable Diffusion 3 Medium). The top row shows evaluation using Image Reward Xu et al. (2023) as the quality metric and Depth Diversity as the diversity metric. The bottom row uses BLIP2 Li et al. (2023) and CLIP Diversity. Note that these metrics are unseen and not used by our method.

as creating more varied and potentially harder-to-detect misleading content, demanding proactive ethical guidelines and mitigation strategies.

## A.5 LLM USAGE

We use LLM, specifically Google Gemini, in the preparation of this paper. Specifically, LLM was used to check for grammar and clarity of writing. LLMs were not used for any ideation, technical content, or analysis.

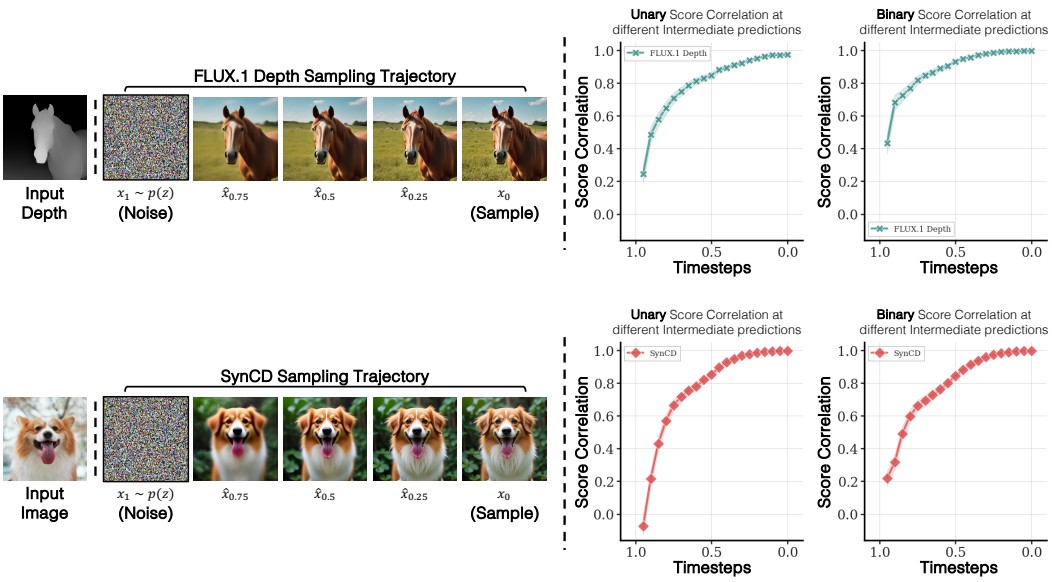

Figure 17: **Correlation between intermediate and final generation Scores .** We follow the same protocol as Figure 3 in the main paper. On the left, we show the reverse diffusion process, visualizing the intermediate predictions $\hat{\mathbf{x}}_t$ of the final image at different steps for FLUX.1 Depth and SynCD models. We can observe that the intermediate predictions look similar to true final sample $\mathbf{x}_0$ for both the models. We further demonstrate this quantitatively by plotting the Spearman correlation of the Unary and Binary scores from $\hat{\mathbf{x}}_t$ versus final $\mathbf{x}_0$ scores, across different steps.

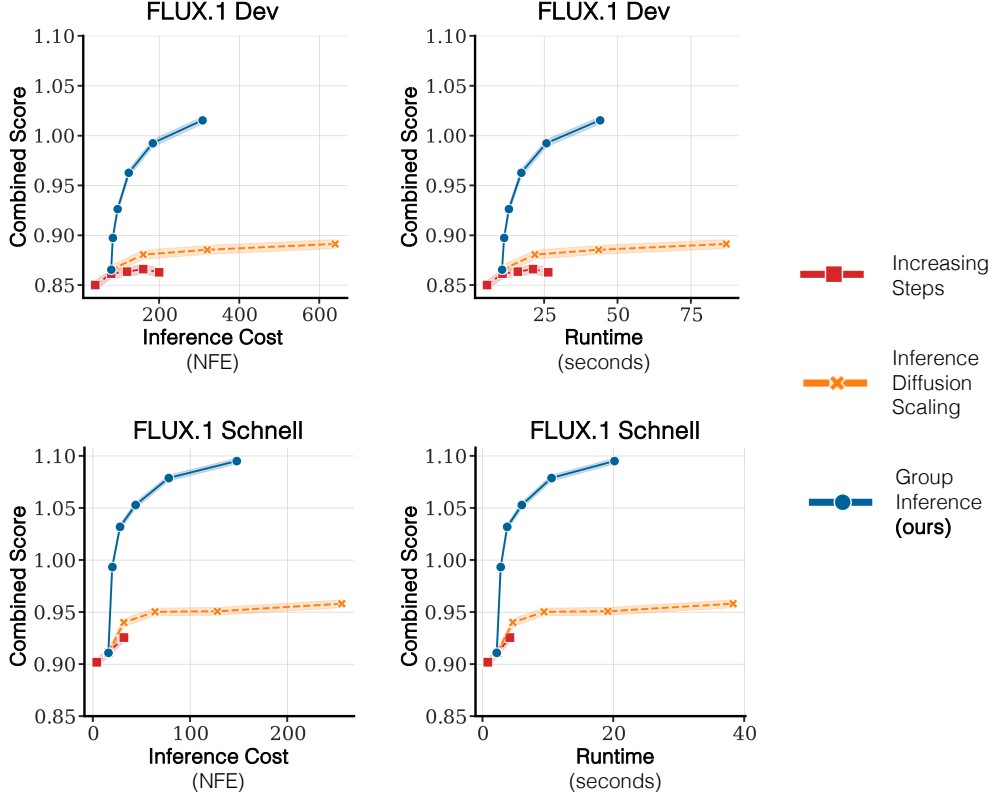

Figure 18: **Performance at different runtimes.** Similar for Figure 6 from the main paper, we show the different ways of allocating inference budget. We report both, the number of function evaluations (NFEs) and the wallclock runtime (using one NVIDIA H100).

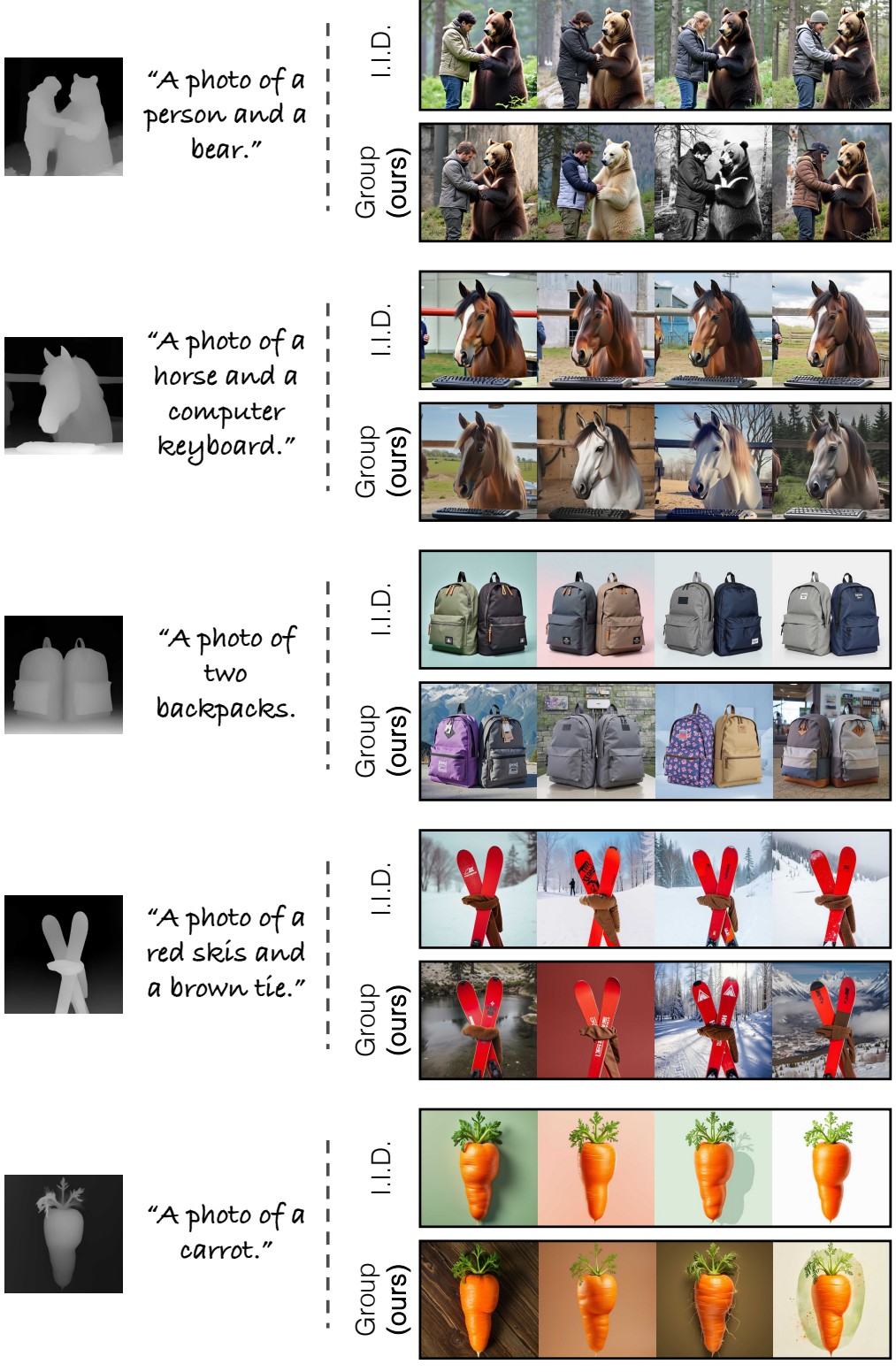

Figure 19: **Gallery of results.** Qualitative results that show the advantage of our proposed method over I.I.D. sampling for depth-to-image generation using FLUX.1 Depth as the base model. The input depth maps and captions are shown on the left and the generated outputs are shown on the right. Our method consistently generates outputs that have more diverse backgrounds, styles, and textures.

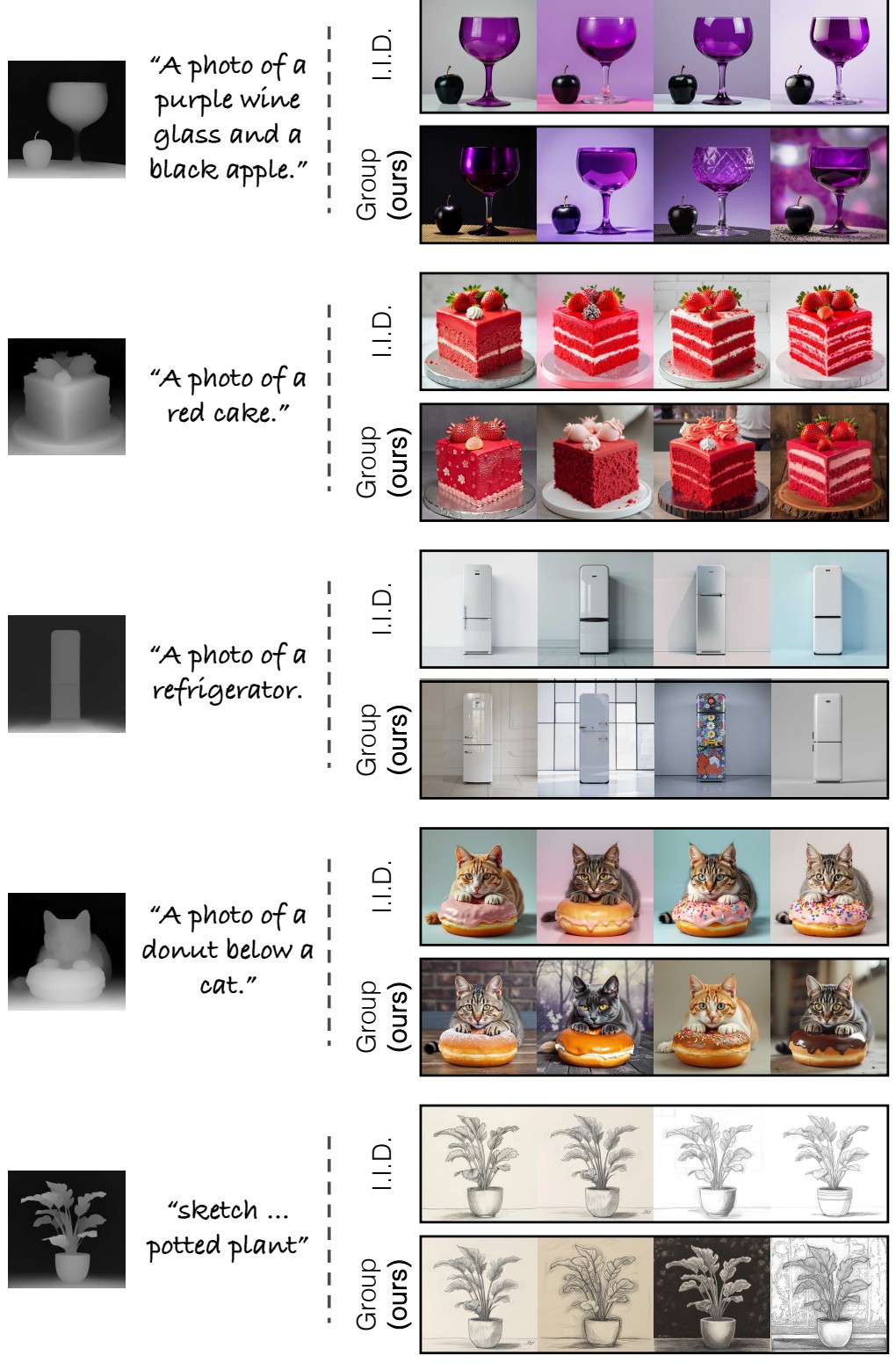

Figure 20: **Gallery of results.** Qualitative results that show the advantage of our proposed method over I.I.D. sampling for depth-to-image generation using FLUX.1 Depth as the base model. The input depth maps and captions are shown on the left and the generated outputs are shown on the right. Our method consistently generates outputs that have more diverse backgrounds, styles, and textures.

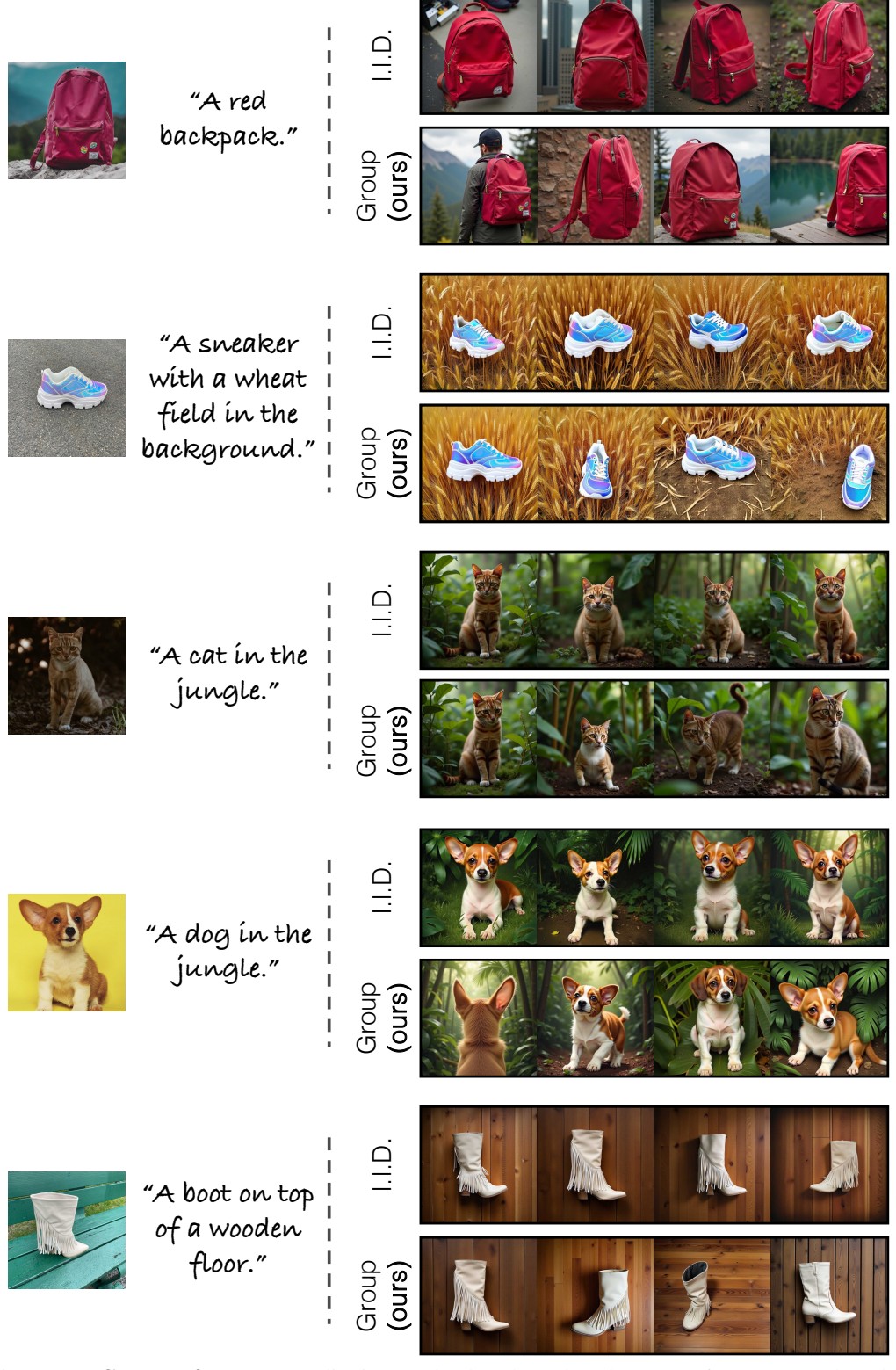

Figure 21: **Gallery of results.** Qualitative results that show the advantage of our proposed method over I.I.D. sampling for feedforward customized generation using SynCD Kumari et al. (2025). The input image and captions are shown on the left and the generated outputs are shown on the right. Our method consistently generates outputs that have more diverse backgrounds, object poses, and styles.

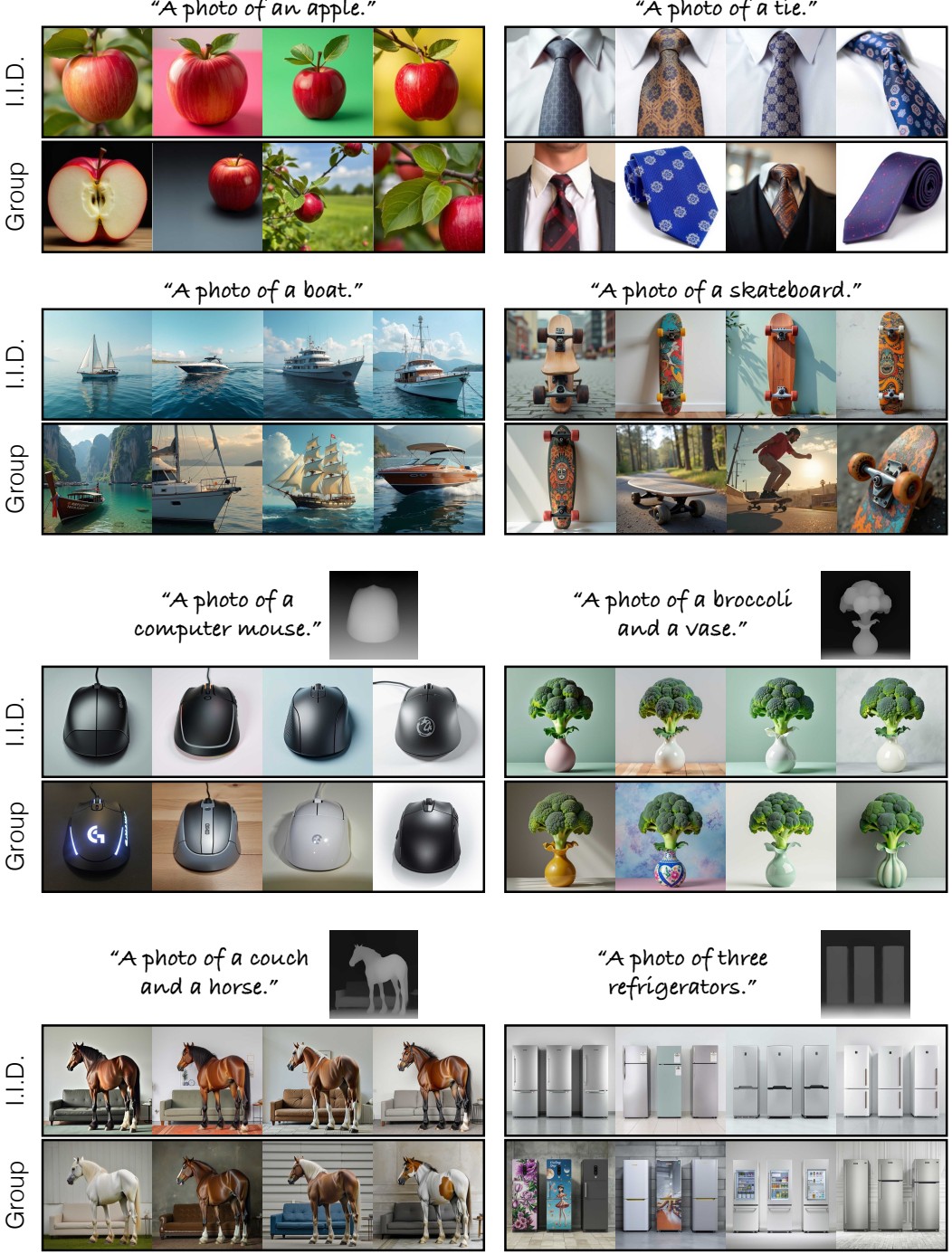

Figure 22: **Gallery of Results.** Qualitative results that show the advantage of our proposed group inference method over I.I.D. sampling for text-to-image generation and depth-to-image generation. Top row shows results with FLUX.1 Schnell, the second row uses FLUX.1 Dev, and the last two rows use FLUX.1 Depth as the base model. For text-to-image generation, our method produces more diverse object poses and orientations, while for depth-to-image generation, it enhances color and texture diversity while adhering to the input depth condition.

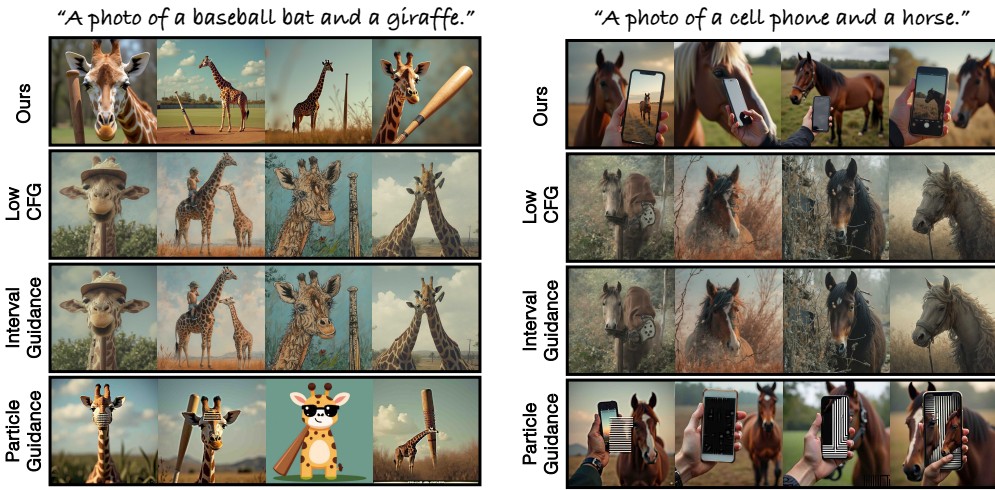

Figure 23: **Qualitative results.** We compare our proposed method (top row) against alternative inference strategies targeting an improved Quality-Diversity tradeoff with FLUX.1 Dev base model.

