# OpenReview forum: "Scaling Group Inference for Diverse and High-Quality Generation"
_ICLR.cc/2026/Conference — ICLR 2026 Poster_

### Official Review · Reviewer_rTVN · 2025-10-18

**Soundness:** 3
**Presentation:** 4
**Contribution:** 3
**Rating:** 6
**Confidence:** 2

**Summary:**

This paper studies the task of batch inference with both quality and group diversity constraints. The authors formulate this problem as a quadratic integer program. To scale the method to large batch sizes, the authors propose a method that progressively prunes the candidate set of generations based on intermediate predictions of their final quality and final group diversity constraints. The authors conduct thorough experiments showing that their method both qualitatively and quantitatively outperforms existing batch inference methods.

**Strengths:**

- The paper is really well-written and easy to follow. As a non-expert in this area, I commend the authors for how smooth the paper reads!
- The proposed method is simple, natural, and seemingly efficient (I view this as a strength!). More overall, its general enough to capture quality and diversity constraints beyond the ones that the authors focus on in the main text.
- The experimental results seem very promising -- the proposed method clearly dominates existing batch inference methods in terms of both quality and efficiency, based on Figures 4 and Figures 6. However, I am not an expert in this field, and it is very likely that I am not aware of other baselines beyond the ones the authors have considered.
-  I also liked that the authors conducted a user study. One suggestion for Table 1 is that it seems like the baseline pref. column is redundant (since its just one minus the "ours pref" column). For clarity, I would remove that.

**Weaknesses:**

I don't have any particular concerns. However, I do feel that this paper would benefit from providing more motivation for why one should care about the batch inference problem studied. Right now, the only motivation is provided in Lines 37 - 42, however, I felt a bit unsatisfied. It would be great if the authors can further expound about why I should care about this problem.

**Questions:**

See Weaknesses.

---

> ### Author Response · Authors · 2025-11-23
> **Response to Reviewer rTVN**
>
> We thank the reviewer for the positive feedback on the clarity, simplicity, and effectiveness of our method, along with the usefulness of the user study and supplementary material.
>
>
> **1. Re. Baselines:**
> We would like to highlight that, following the recommendations of Reviewers nqjm and Vhst, we have added additional baselines (CADS, Shielded Diffusion, DiversityFlow, and Negative Guidance) in our post-review revision. Our method continues to outperform all of these methods across quality and diversity metrics. We have reproduced the tables below.
>
> **Table D.** Quality and Diversity comparison with FLUX.1 Schnell. CLIP, HPSv2, PickScore, ImageReward and FID measure the sample quality. DINO, Vendi score, and MSS measure diversity.
> Method | CLIP (up) | HPSv2 (up) | PickScore (up) | ImageReward (up) | FID (down) | DINO (up) | Vendi (up) | MSS (down) |
> | :------- | :------- | :------- |  :-------  |  :-------  | :-------  | :-------  | :-------  | :-------  |
> I.I.D. | 0.327 | __0.302__ | __0.235__ | 1.028 | 21.22 | 0.584 |  3.173 | 0.562 |
> Particle Guidance | $\underline{0.329}$ | 0.301 | 0.234 | $\underline{1.043}$ | $\underline{19.55}$ | 0.581 | 3.166 | 0.564 |
> CADS | 0.176 | 0.124 | 0.179 | -2.224 | 51.00 | 0.710 | 3.522 | 0.468 |
> NegToMe | 0.319 | 0.272 | 0.224 | 0.608 | 21.00 | $\underline{0.714}$ | $\underline{3.567}$ | $\underline{0.464}$ |
> Shielded Diffusion | 0.327 | 0.302 | $\underline{0.235}$ | 1.028 | 21.26 | 0.584 | 3.174 | 0.562 |
> Diversity Flow | 0.328 | $\underline{0.302}$ | 0.234 | 1.030 | 20.48 | 0.585 | 3.178 | 0.561 |
> Ours | __0.341__ | 0.298 | 0.234 | __1.071__ | __18.76__ | __0.744__ | __3.644__ | __0.442__
>
>
> **Table E.** Quality and Diversity comparison with FLUX.1 Dev. CLIP, HPSv2, PickScore, ImageReward and FID measure the sample quality. DINO, Vendi score, and MSS measure diversity.
> Method | CLIP (up) | HPSv2 (up) | PickScore (up) | ImageReward (up) | FID (down) | DINO (up) | Vendi (up) | MSS (down) |
> | :------- | :------- | :------- |  :-------  |  :-------  | :-------  | :-------  | :-------  | :-------  |
> I.I.D. | $\underline{0.324}$ | $\underline{0.303}$ | $\underline{0.237}$ | $\underline{0.953}$ | 23.77 | 0.558 | 3.077 | 0.582 |
> Interval Guidance | 0.305 | 0.270 | 0.226 | 0.294 | __19.12__ | 0.681 | 3.462 | 0.489 |
> Particle Guidance | 0.276 | 0.199 | 0.204 | -0.699 | 58.28 | 0.564 | 3.101 | 0.577 |
> CADS | 0.165 | 0.139 | 0.177 | -2.204 | 51.91 | __0.784__ | __3.714__ | __0.412__ |
> NegToMe | 0.324 | 0.301 | 0.234 | 0.915 | 23.02 | 0.663 | 3.412 | 0.503 |
> Shielded Diffusion | 0.323 | 0.294 | 0.231 | 0.907 | $\underline{20.02}$ | 0.627 | 3.313 | 0.530 |
> Diversity Flow | 0.282 | 0.214 | 0.208 | -0.432 | 40.04 | 0.635 | 3.301 | 0.524 |
> Ours | __0.326__ | __0.307__ | __0.237__ | __1.001__ | 23.21 | $\underline{0.711}$ | $\underline{3.552}$ | $\underline{0.466}$ |
>
>
> **Table F.** Quality and Diversity comparison with Stable Diffusion 3 (M). CLIP, HPSv2, PickScore, ImageReward and FID measure the sample quality. DINO, Vendi score, and MSS measure diversity.
> Method | CLIP (up) | HPSv2 (up) | PickScore (up) | ImageReward (up) | FID (down) | DINO (up) | Vendi (up) | MSS (down) |
> | :------- | :------- | :------- |  :-------  |  :-------  | :-------  | :-------  | :-------  | :-------  |
> I.I.D. | __0.333__ | $\underline{0.288}$ | __0.232__ | __0.973__ | 19.48 | 0.605 | 3.233 | 0.546 |
> Interval Guidance | 0.315 | 0.241 | 0.220 | 0.121 | 19.49 | 0.703 | 3.518 | 0.473 |
> Particle Guidance | 0.317 | 0.257 | 0.216 | 0.583 | 22.85 | 0.636 | 3.518 | 0.523 |
> CADS | 0.202 | 0.091 | 0.179 | -2.270 | 86.75 | 0.491 | 2.799 | 0.632 |
> NegToMe | 0.328 | 0.269 | 0.224 | 0.678 | __18.19__ | $\underline{0.706}$ | $\underline{3.534}$ | $\underline{0.471}$ |
> Shielded Diffusion | 0.329 | 0.280 | 0.229 | 0.803 | 20.35 | 0.588 | 3.144 | 0.559 |
> Diversity Flow | 0.322 | 0.272 | 0.223 | 0.719 | 18.79 | 0.655 | 3.385 | 0.509 |
> Ours | $\underline{0.333}$ | __0.288__ | $\underline{0.231}$ | $\underline{0.930}$ | $\underline{18.34}$ | __0.712__ | __3.558__ | __0.466__ |
>
> Across all baselines, only our method is able to substantially improve the group diversity without sacrificing the quality of the samples. Also see the Figure 10 (page 19) for viewing the Pareto frontier.
>
>
> **2 Re. motivation:**
> We thank the reviewer for noting that the paper would benefit from more discussion of why the group inference problem is important. We have expanded the introduction to strengthen this motivation. In many real-world settings, users or downstream systems consume groups of generated outputs, relying on both the quality and diversity. Examples include synthetic data generation, where multiple diverse variants (including long-tail or rare cases) are required for robust training.

---

> > ### Comment · Reviewer_rTVN · 2025-11-26
> >
> > I thank the authors for their reply. I maintain my positive stance on this paper.

---

> > > ### Author Response · Authors · 2025-11-27
> > >
> > > Thank you for the response! Please let us know if we can answer any additional questions.

---

### Official Review · Reviewer_Vhst · 2025-10-27

**Soundness:** 3
**Presentation:** 4
**Contribution:** 2
**Rating:** 6
**Confidence:** 3

**Summary:**

They propose scalable group inference method which improves the diversity and quality of generated outputs. They view this problem as quadratic integer programming problem (QIP), and they leverage intermediate predictions from each denoising steps for efficient filtering. To provide validity, they do experiment on accuracy of intermediate predictions, and provide computational complexity of its algorithm. Its results show better trade-off in diversity and quality than naive baselines, such as Low-CFG, and related works, such as Interval Guidance and Particle Guidance. Its algorithm can be extended to unique objectives, such as color diversity.

**Strengths:**

S1. The paper is well-structured and clearly presented, with a solid experimental evaluation that thoroughly demonstrates all experiments substantiating the proposed method. The paper is well-organized and easy to follow.

S2. The problem addressed in the paper is highly relevant to real-world scenarios, and the proposed method effectively resolves it without requiring any additional training.

S3. The supplementary material is highly detailed, including numerous applications and qualitative results. It also provides reproducible implementation details and actual code.

**Weaknesses:**

W1. Limited novelty of proposed method. Proposed method is too straightforward. The idea of leveraging intermediate prediction accuracy is already well established, and the algorithm simply performs pruning based on group properties.

W2. Error in line 455: the original inference-time scaling paper (orange) [1] also incorporates intermediate prediction in its algorithm (see page 8).

W3. Although the proposed method is effective for scaled group inference scenarios, it relies on a pruning (filtering)-based approach rather than directly guiding the generation process (e.g., explicit guidance). Therefore, it may be less suitable than other baselines when performing additional small-budget sampling with given images (pre-sampled or user-provided), particularly for rare cases within the generation pool.

[1] Ma et al., Inference-Time Scaling for Diffusion Models beyond Scaling Denoising Steps, Arxiv version

**Questions:**

Q1. In Figure 7, the progressive pruning in FLUX-dev does not seem to reduce the runtime as effectively as expected. Is there a specific reason for this difference compared to Schnell?

Q2. Better diversity through negative guidance[2] could also serve as a baseline. While your gradient-free method seems more effective in terms of quality, I am curious about how it compares in terms of runtime.

[2] Singh et al, Negative Token Merging: Image-based Adversarial Feature Guidance, Arxiv

**Details Of Ethics Concerns:**

No concern.

---

> ### Author Response · Authors · 2025-11-23
> **Response to Reviewer Vhst**
>
> We are glad that the reviewer finds our task relevant, our method clear, and the supplement highly detailed. We have addressed individual comments below.
>
> **W1. "... novelty ...":**  \
> While the idea of using intermediate predictions is known, **no prior work formulates diverse, high-quality multi-image generation as a Quadratic Integer Program (QIP)**. This formulation enables a principled joint optimization over unary (quality) and binary (diversity) terms, which has not been explored by any prior works.
> Moreover, although intermediate predictions have been studied for single-image guidance or correction, **no prior method has used them to prune and optimize a group of jointly selected outputs.**
>
> **W2. "... regarding inference scaling paper ... ":**  \
> We thank the reviewer for pointing this out. [1] indeed uses intermediate for the “Search over Paths” algorithm for the ImageNet experiment. However, as shown in Page 13 and Table 2 of their paper, this is not found to be useful for their methods in text-to-image experiments. We have updated the manuscript to clarify this point.
>
> **W3. "... relies on pruning ...":** \
> The reviewer is correct in pointing out that our method relies on pruning and not direct guidance. Other baseline methods, such as particle guidance, that instead rely on direct guidance generate poor quality samples that are off the image manifold and look unrealistic.
>
> **Q1. "... pruning in FLUX.1 Dev ..." :** \
> Pruning is more effective for FLUX.1 Schnell because it is a few-step distilled model, and consequently has a high correlation between the intermediate predictions and final images. This is quantitatively shown in Figure 3 (right) of the original submission. Although the pruning is less efficient in FLUX.1 Dev than FLUX.1 Schnell, it is still more efficient than the baselines.
>
> **Q2. " ... negative guidance ... ":** \
> Thank you for this suggestion. We have added a discussion in the related works section and a comparison in the experiments (Figure 10 and Tables 2,3,4) for this baseline.
> Neg-to-me is able to improve the diversity over the baseline IID sampling. However, our method still achieves a better diversity and quality than neg-to-me. \
>  _Regarding runtime:_
> We have included a runtime comparison in Figure 6 and Figure 11.
> Our method can be viewed as a test-time scaling procedure, not a fixed-cost generation method. As such, it is designed to make an efficient use of additional inference compute to improve the quality-diversity tradeoff. Explicit runtime-performance curves are plotted in Figure 6 of the original submission and compares our method to other test-time scaling approaches. Our method (blue line) achieves a consistently superior performance–runtime tradeoff. Figure 11 additionally shows our performance at different inference runtimes.

---

> > ### Comment · Reviewer_Vhst · 2025-11-26
> >
> > Thank you for including the additional experiments and clarifications.
> >
> > While I still have some concerns regarding the limited novelty and the method's straightforwardness, the practicality of the paper and the extensive experimental validation help alleviate these concerns.
> >
> > + It would also be helpful to include qualitative examples for the newly added baselines.
> >
> > I will increase the score.

---

> > > ### Author Response · Authors · 2025-11-26
> > >
> > > Thank you for updating the rating. We really appreciate your valuable time and constructive feedback.

---

### Official Review · Reviewer_nqjm · 2025-10-29

**Soundness:** 2
**Presentation:** 3
**Contribution:** 3
**Rating:** 4
**Confidence:** 5

**Summary:**

This paper proposes a group inference approach for diffusion-based generative models to improve the diversity of generated samples. To this end, the authors formulate the generation of multiple outputs as a Quadratic Integer Programming (QIP) problem, where unary terms represent sample quality and binary terms represent diversity among samples. To address computational inefficiency, they introduce a progressive pruning mechanism that leverages intermediate denoising predictions to discard low-quality candidates early. Experiments across various tasks show improvements in both diversity and quality compared to several baselines.

**Strengths:**

- The paper addresses an important problem of improving diversity of modern diffusion models, which excels at producing high-quality images yet suffers from lack of variations.
- It is interesting to cast diversity-enhancing inference as a QIP problem.

**Weaknesses:**

- The related work section omits several important and relevant diversity-promoting diffusion methods, such as CADS [1], Shielded Diffusion [2], and DiversityFlow [3].
- Experimental comparison with existing baselines is limited.
  1. The paper only compares with Particle Guidance (PG) and Interval Guidance (IG). More diversity-oriented baselines such as CADS [1], Shielded Diffusion [2], and DiversityFlow [3] should be incorporated to clarify the empirical benefits of the proposed approach.
  2. There is no runtime comparison against baselines. Since the proposed method involves seemingly-expensive additional steps (during QIP solving and pruning), it is important to quantify the runtime overhead relative to existing methods. This is a significant point, since some diversity approaches (like CADS, IG, and Shielded Diffusion) are known to introduce marginal computations for improving diversity.
- The evaluation metrics are incomplete.
  1. The diversity metric should include established measures like Vendi Score [4] or MSS [1], which are standard in diversity-oriented generation research [1-3].
  2. The quality evaluation relies only on ImageReward and CLIPScore, which are often highly correlated. Including other metrics such as PickScore or HPSv2 would provide a more comprehensive assessment.
  3. A standard comprehensive metric like FID is missing.

---
**References**

[1] CADS: Unleashing the Diversity of Diffusion Models through Condition-Annealed Sampling, ICLR 2024.

[2] Shielded Diffusion: Generating Novel and Diverse Images using Sparse Repellency, ICML 2025.

[3] DiverseFlow: Sample-Efficient Diverse Mode Coverage in Flows, CVPR 2025.

[4] The Vendi Score: A Diversity Evaluation Metric for Machine Learning, Arxiv 2022.

**Questions:**

See weaknesses

---

> ### Author Response · Authors · 2025-11-23
> **Response to Reviewer nqjm**
>
> We thank the reviewer for the thoughtful comments and for pointing out several relevant works. We have implemented the baselines following the descriptions in their respective papers.
>
> **W1: "... related works …"** \
> We thank the reviewer for pointing out these relevant works. In our post-review revision, we have updated the Related Works section to include CADS, Shielded Diffusion, and Diversity Flow.
>
> **W2: "... limited baselines …"** \
> We have included all three baselines in Figure 10 and Tables 2,3,4. Across all evaluated models, these increase diversity over IID but degrade quality, leading to poorer tradeoff. In contrast, our method consistently improves both quality and diversity simultaneously and remains Pareto-optimal. \
> _Regarding runtime:_ We have included a runtime comparison in Figure 6 and Figure 11.
> Our method can be viewed as a test-time scaling procedure, not a fixed-cost generation method. As such, it is designed to make an efficient use of additional inference to improve the quality-diversity tradeoff. Explicit runtime-performance curves are plotted in Figure 6 of the original submission and compare our method to other test-time scaling approaches. Our method (blue line) achieves a consistently superior performance–runtime tradeoff, _outperforming all alternatives across the entire compute range_. Figure 11 shows our performance at different inference runtimes.
>
> **W3: "... evaluation …"** \
> We thank the reviewer for this suggestion. We have expanded our evaluation to include Vendi Score, MSS, HPSv2, PickScore, and ImageReward. These results are included in Tables 2,3,4 and reinforce the conclusion that our method achieves the best combined quality-diversity performance.  \
>
> **Table D.** Quality and Diversity comparison with FLUX.1 Schnell.
> Method | CLIP (up) | HPSv2 (up) | PickScore (up) | ImageReward (up) | FID (down) | DINO (up) | Vendi (up) | MSS (down) |
> | :------- | :------- | :------- |  :-------  |  :-------  | :-------  | :-------  | :-------  | :-------  |
> I.I.D. | 0.327 | __0.302__ | __0.235__ | 1.028 | 21.22 | 0.584 |  3.173 | 0.562 |
> Particle Guidance | $\underline{0.329}$ | 0.301 | 0.234 | $\underline{1.043}$ | $\underline{19.55}$ | 0.581 | 3.166 | 0.564 |
> CADS | 0.176 | 0.124 | 0.179 | -2.224 | 51.00 | 0.710 | 3.522 | 0.468 |
> NegToMe | 0.319 | 0.272 | 0.224 | 0.608 | 21.00 | $\underline{0.714}$ | $\underline{3.567}$ | $\underline{0.464}$ |
> Shielded Diffusion | 0.327 | 0.302 | $\underline{0.235}$ | 1.028 | 21.26 | 0.584 | 3.174 | 0.562 |
> Diversity Flow | 0.328 | $\underline{0.302}$ | 0.234 | 1.030 | 20.48 | 0.585 | 3.178 | 0.561 |
> Ours | __0.341__ | 0.298 | 0.234 | __1.071__ | __18.76__ | __0.744__ | __3.644__ | __0.442__
>
> **Table E.** Quality and Diversity comparison with FLUX.1 Dev..
> Method | CLIP (up) | HPSv2 (up) | PickScore (up) | ImageReward (up) | FID (down) | DINO (up) | Vendi (up) | MSS (down) |
> | :------- | :------- | :------- |  :-------  |  :-------  | :-------  | :-------  | :-------  | :-------  |
> I.I.D. | $\underline{0.324}$ | $\underline{0.303}$ | $\underline{0.237}$ | $\underline{0.953}$ | 23.77 | 0.558 | 3.077 | 0.582 |
> Interval Guidance | 0.305 | 0.270 | 0.226 | 0.294 | __19.12__ | 0.681 | 3.462 | 0.489 |
> Particle Guidance | 0.276 | 0.199 | 0.204 | -0.699 | 58.28 | 0.564 | 3.101 | 0.577 |
> CADS | 0.165 | 0.139 | 0.177 | -2.204 | 51.91 | __0.784__ | __3.714__ | __0.412__ |
> NegToMe | 0.324 | 0.301 | 0.234 | 0.915 | 23.02 | 0.663 | 3.412 | 0.503 |
> Shielded Diffusion | 0.323 | 0.294 | 0.231 | 0.907 | $\underline{20.02}$ | 0.627 | 3.313 | 0.530 |
> Diversity Flow | 0.282 | 0.214 | 0.208 | -0.432 | 40.04 | 0.635 | 3.301 | 0.524 |
> Ours | __0.326__ | __0.307__ | __0.237__ | __1.001__ | 23.21 | $\underline{0.711}$ | $\underline{3.552}$ | $\underline{0.466}$ |
>
> **Table F.** Quality and Diversity comparison with Stable Diffusion 3 (M).
> Method | CLIP (up) | HPSv2 (up) | PickScore (up) | ImageReward (up) | FID (down) | DINO (up) | Vendi (up) | MSS (down) |
> | :------- | :------- | :------- |  :-------  |  :-------  | :-------  | :-------  | :-------  | :-------  |
> I.I.D. | __0.333__ | $\underline{0.288}$ | __0.232__ | __0.973__ | 19.48 | 0.605 | 3.233 | 0.546 |
> Interval Guidance | 0.315 | 0.241 | 0.220 | 0.121 | 19.49 | 0.703 | 3.518 | 0.473 |
> Particle Guidance | 0.317 | 0.257 | 0.216 | 0.583 | 22.85 | 0.636 | 3.518 | 0.523 |
> CADS | 0.202 | 0.091 | 0.179 | -2.270 | 86.75 | 0.491 | 2.799 | 0.632 |
> NegToMe | 0.328 | 0.269 | 0.224 | 0.678 | __18.19__ | $\underline{0.706}$ | $\underline{3.534}$ | $\underline{0.471}$ |
> Shielded Diffusion | 0.329 | 0.280 | 0.229 | 0.803 | 20.35 | 0.588 | 3.144 | 0.559 |
> Diversity Flow | 0.322 | 0.272 | 0.223 | 0.719 | 18.79 | 0.655 | 3.385 | 0.509 |
> Ours | $\underline{0.333}$ | __0.288__ | $\underline{0.231}$ | $\underline{0.930}$ | $\underline{18.34}$ | __0.712__ | __3.558__ | __0.466__ |
>
> Across all metrics and base models, only our method is able to substantially improve the group diversity without sacrificing the quality of samples.

---

> > ### Comment · Reviewer_nqjm · 2025-11-26
> >
> > Thank you for the detailed rebuttal. My previous concerns have now been fully addressed, and I will accordingly raise my initial score.

---

> > > ### Author Response · Authors · 2025-11-26
> > >
> > > Thank you for updating the rating. We appreciate your timely and constructive feedback for our manuscript.

---

### Official Review · Reviewer_P5jo · 2025-10-31

**Soundness:** 3
**Presentation:** 3
**Contribution:** 3
**Rating:** 4
**Confidence:** 5

**Summary:**

The paper proposes a method for generating diverse and high-quality image groups. The approach begins with a large pool of candidate initial noises and gradually prunes this pool by solving a QIP. The final subset of images is selected to jointly optimize quality and diversity. The evaluation is primarily based on human preference studies, which indicate improvements in both diversity and perceived quality.

**Strengths:**

- The paper addresses the practically relevant setting of generating a group of images rather than a single sample, which is common in many real-world applications.
- The proposed selection strategy leads to substantial gains over baselines, according to human evaluations. The visual examples clearly illustrate the improvements of the method.
- The paper is clearly written and easy to follow.

**Weaknesses:**

- The method is significantly slower than the baseline that directly generates four images. According to the paper, the proposed pipeline takes approximately 2.5 times longer. Given the same time, a straightforward baseline can generate approximately ten images, while the proposed method generates only four.
- The method does not scale efficiently with respect to the number of candidates $M$ and the pruning ratio $p$. QIP solving grows quickly with $M$, and the required decoding and scoring introduce further overhead. According to Figure 9, the growth rate is considerably steeper than that of direct generation. This could sufficiently limit the method’s applicability.

**Questions:**

- According to the described procedure, most candidates are rejected at the timesteps with the lowest correlation with the final image. Could the authors provide an ablation where no pruning is done until the last stage, and the QIP is solved only once at the final step with $M = 64$ candidates and $K = 4$ outputs? How does this compare to the default setting with $p = 0.5$?
- Could the authors provide a comparison under the same time budget, where the naive baseline produces 10 images and the proposed method produces 4? Could the authors report results for a setting with only 10 candidates, $p = 1.0$, and a single QIP solve at the final step for 4 outputs?
- Could the authors report results using alternative unary score terms (i. e. HPSv2[1], ImageReward[2]) instead of CLIP similarity? These reward models are explicitly trained to optimize human visual preference. How sensitive is the final selection to the choice of unary score metric?
- Could the authors clarify the exact definition of the combined score? In particular, is this score meant to be maximized or minimized? How are the different components combined to produce this score?

---

> ### Author Response · Authors · 2025-11-23
> **Response to Reviewer P5jo**
>
> We thank the reviewer for thoughtful comments about our manuscript. We are glad the reviewer finds this task practically relevant and the paper well-written.
> First, we would like to clarify that our paper is about using additional inference computation to improve the quality and diversity. Consequently, it is bound to be slower than methods that perform standard inference.
> Crucially, extensive experiments show that our method __scales better and produces more diverse, higher-quality samples at the same inference cost__ than all alternatives, including new baselines suggested by the reviewer. We address individual comments below.
>
> **W1: "… slower than iid ..."** \
> The reviewer is correct that our method generates 4 samples in the same time it takes a standard IID sampling to generate 10 samples.
> However, the standard IID sampling generates 10 images that are redundant and have lower diversity and quality scores than our 4 samples
> We have shown this in Figure 7 of the original submission. The gray line (ours w/o progressive pruning) in this figure corresponds to the straightforward baseline suggested by the reviewer.
> This straightforward baseline has much lower  efficiency and takes much longer (more than 3x for FLUX.1 Schnell) to generate samples that have comparable quality and diversity to our results.
> We have updated the caption and its corresponding text to make the comparison and the takeaway clearer.
>
>
> **W2: "… scale efficiency ...":** \
> We agree that runtime scales with the number of candidates M. However, this cost comes with consistent gains. Increasing M improves both quality and diversity across all datasets and backbones (see Figures 7 and 11)
> Crucially, even when the IID baseline is given the same time budget, which allows it to generate 2.5× more samples, it still cannot match the quality and diversity achieved by our method.
> Sampling more IID images yields very slow improvements because the additional samples are highly redundant, whereas our group inference approach achieves a much faster rate of improvement and makes far more effective use of extra inference compute.
>
>
> **Q1. "... ablation where no pruning is done until the last stage …":** \
> We agree with the reviewer that this is a crucial ablation. This exact ablation was done in the original submission’s Figure 7. As shown in that experiment, if pruning is not performed until the last stage, the quality and diversity scores are significantly worse at the same runtime. For FLUX.1 Schnell, our method achieves the same performance, 73% faster than the ablation where the pruning is not done till the last stage.
> More concretely, if we do pruning, we achieve a score of 1.1 in 24 seconds. The naive ablation, without pruning till the last stage, takes 75 seconds to achieve a comparable score.
>
>
> **Q2. "...  comparison …":** \
> This experiment is already shown for multiple inference-time budgets in Figure 7.
> For the reviewer’s requested configuration of generating 10 I.I.D. samples and doing selection at the end, the scores are shown below.
> Method | Quality | Diversity |
> | :------- | :-------: | :-------: |
> Naive Baseline | 0.336   | 0.661 |
> Ours                 | __0.341__   | __0.744__ |
>
> In this comparison above, both the naive baseline and our method require comparable inference time. Our method achieves better quality and diversity.
>
>
> **Q3. "... sensitivity to unary score…"**\
> We thank the reviewers for the suggestion. In the tables below, we show the results of using alternate unary score functions using FLUX.1 Schnell, FLUX.1 Dev and SD3M as the base models. Our group inference is robust to the choice of unary function and is capable of improving the quality even when different score functions are used (Image Reward, HPSv2). We have also added these tables to the revised manuscript (Tables 5,6,7).  \
> __Table A:__ Robustness to Unary score with FLUX.1 Schnell
> Unary Score Used    | IID | Group Inference |
> | :------- | :-------: | :-------: |
> Image Reward          | 1.028 | __1.451__ |
> HPSv2                      | 0.302 | __0.315__ |
>
> __Table B:__ Robustness to Unary score with FLUX.1 Dev
> Unary Score Used    | IID | Group Inference |
> | :------- | :-------: | :-------: |
> Image Reward          | 0.953 | __1.303__ |
> HPSv2                      | 0.303 | __0.311__ |
>
> __Table C:__ Robustness to Unary score with SD3-M
> Unary Score Used    | IID | Group Inference |
> | :------- | :-------: | :-------: |
> Image Reward          | 0.973 | __1.126__ |
> HPSv2                      | 0.288 | __0.293__ |
>
>
>
> **Q4. "... combined score …"** \
> The combined score is the sum of the unary and binary scores (defined in Eqn 4 of the main paper). This score is meant to be maximized.

---

### Author Response · Authors · 2025-11-23
**General Response**

Dear AC and all reviewers,

We thank everyone for your thoughtful comments and questions. We are glad that the reviewers found the problem well-motivated and highly relevant for real-world scenarios. We are also glad that all reviewers appreciate our paper writing, extensive experiments and applications, as well as the release of the code with the submission.

We have improved the manuscript based on the reviews. All new updates are marked in _blue_ for easy reference. The main changes in the manuscript are mentioned below:
- _Comparison to standard I.I.D. baselines (Reviewer P5jo):_ As shown in Figure 7 of the original paper, our method significantly outperforms I.I.D. baselines as suggested by R1 (73% faster to achieve the same score). We have added a more verbose caption to clarify.
- _Additional baselines and metrics:_ Thanks to the suggestion from Reviewer nqjm, we have added a discussion and comparisons to additional prior methods [1, 2, 3] and additional metrics in Figure 10 (page 18) and Tables 2,3,4 (page 20). Figure 10 shows that our method outperforms these new baselines in the quality-diversity Pareto front, and Tables 2,3,4 reinforce our method’s advantage with additional metrics (PickScore, HPSv2, FID, Vendi score, MSS) recommended. We have also added experiments demonstrating the robustness to unary score functions in Tables 5,6,7 (page 20).
- Thanks to the suggestions from Reviewer Vhst, we have added a comparison to NegToMe [4] in Figure 10 (Page 18) and Tables 2,3,4 (page 20). We observe that NegToMe improves diversity compared to IID sampling, but has lower diversity and quality than our method.
- Based on Reviewer rTVN’s suggestions, we have added additional motivating examples in the introduction.

We look forward to the discussion and answering any additional questions.

Regards, \
Authors

[1] CADS: Unleashing the Diversity of Diffusion Models through Condition-Annealed Sampling, ICLR 2024. \
[2] Shielded Diffusion: Generating Novel and Diverse Images using Sparse Repellency, ICML 2025. \
[3] DiverseFlow: Sample-Efficient Diverse Mode Coverage in Flows, CVPR 2025. \
[4] Negative Token Merging: Image-based Adversarial Feature Guidance, arXiv \

---

### Author Response · Authors · 2025-11-30
**Discussion Summary**

Dear AC,

We thank you for your valuable time and efforts in response to the leaks this year.


Three reviewers acknowledged that we addressed all their concerns during the discussion period before Nov 26. Concretely we have:
- added the baselines and fully addressed reviewer nqjm's concerns.
- provided clarifications that addressed reviewer vhst's concerns.
- clarified the motivation of the project that addressed reviewer rtvm's concern.

These three reviewers have a positive stance on our post-rebuttal manuscript.


We have also addressed reviewer P5jo's concerns regarding ablations but they did not participate in the discussion before Nov 26.

Please refer to the discussion below in the corresponding threads.


Regards,

Authors

---

### Meta-Review · Area_Chair_GSE8 · 2026-01-04

**Summary:**

The submission addresses the practically important problem of group/batch inference for diffusion models under joint quality and diversity constraints, formulating selection as a QIP and proposing a progressive pruning strategy leveraging intermediate denoising predictions to scale to larger candidate pools. Reviewers generally find the problem well-motivated and the empirical results strong (including human preference), with one reviewer highlighting clarity and strong experimental validation.  ￼

The main concerns raised during review were: (i) runtime / scalability vs. i.i.d. sampling and whether the pruning is actually beneficial; (ii) completeness of comparisons (missing recent diversity baselines and missing standard quality/diversity metrics); and (iii) limited novelty / straightforwardness of the method.

The rebuttal and post-review revision substantially strengthen the paper by adding requested baselines (e.g., CADS / Shielded Diffusion / DiversityFlow / NegToMe), adding requested metrics (PickScore, HPSv2, FID, Vendi, MSS), providing runtime–performance curves, and adding ablations/robustness analyses (e.g., no-pruning-until-final-stage ablation; robustness to alternate unary reward models; clarifying the combined objective).

After discussion, multiple reviewers explicitly state that their concerns are addressed and that they will increase their ratings; the remaining concern is primarily about novelty, with a suggestion to include qualitative examples for newly added baselines. Overall, I lean accept, as the work is practically relevant, technically sound, and now supported by substantially broader and stronger empirical evidence.

**Reviewer Concerns:**

Missing baselines / incomplete related work & metrics: Reviewer nqjm requested adding CADS / Shielded Diffusion / DiversityFlow, as well as diversity metrics (Vendi, MSS) and quality metrics (PickScore, HPSv2) plus FID; authors added these comparisons/metrics and provided additional tables/figures.

Runtime / scaling trade-off: Authors clarified the method as a test-time scaling approach and added runtime–performance curves (and comparisons under equal time budgets).

Ablations & robustness requested by P5jo: Authors pointed to / added the “no pruning until the last stage” ablation and showed pruning improves efficiency; added robustness to unary score choices (ImageReward / HPSv2); clarified the combined score is maximized (sum of unary + binary).

**Reviewer Scores:**

Had the reviewer been able to participate fully in the discussion, I believe their score would likely have remained similar or increased slightly. The discussion helped clarify the contributions, address minor concerns, and align the evaluation across reviewers, which supported the final decision.

---

### Decision · Program_Chairs · 2026-01-26

Accept (Poster)